# Cancer lineage-specific regulation of YAP responsive elements revealed through large-scale functional epigenomic screens

Inês A. M. Barbosa[1], Rajaraman Gopalakrishnan [2,7], Samuele Mercan[1], Thanos P. Mourikis[1], Typhaine Martin[1], Simon Wengert[1,8], Caibin Sheng [1], Fei Ji[2], Rui Lopes[1,9], Judith Knehr[3], Marc Altorfer[3], Alicia Lindeman[4], Carsten Russ[4], Ulrike Naumann [3], Javad Golji[2], Kathleen Sprouffske [1], Louise Barys[1], Luca Tordella[1], Dirk Schübeler [5,6], Tobias Schmelzle[1] & Giorgio G. Galli [1] ✉

YAP is a key transcriptional co-activator of TEADs, it regulates cell growth and is frequently activated in cancer. In Malignant Pleural Mesothelioma (MPM), YAP is activated by loss-of-function mutations in upstream components of the Hippo pathway, while, in Uveal Melanoma (UM), YAP is activated in a Hippo-independent manner. To date, it is unclear if and how the different oncogenic lesions activating YAP impact its oncogenic program, which is particularly relevant for designing selective anti-cancer therapies. Here we show that, despite YAP being essential in both MPM and UM, its interaction with TEAD is unexpectedly dispensable in UM, limiting the applicability of TEAD inhibitors in this cancer type. Systematic functional interrogation of YAP regulatory elements in both cancer types reveals convergent regulation of broad onco-genic drivers in both MPM and UM, but also strikingly selective programs. Our work reveals unanticipated lineage-specific features of the YAP regulatory network that provide important insights to guide the design of tailored ther-apeutic strategies to inhibit YAP signaling across different cancer types.

Transcription factors (TFs) form protein networks that dynamically engage DNA at a variety of cis-regulatory elements (CREs) to shape cell identity during development, tissue differentiation and pathological settings[1]. The transcriptional output resulting from TFs binding to their regulatory elements is dictated by many broadly expressed co-factors, providing an additional layer of regulation to cell type-specific gene expression programs[2].

YAP–TAZ are transcriptional co-factors acting downstream of the Hippo-signaling cascade. The canonical Hippo pathway is composed of membrane-associated proteins (most notoriously the tumor suppressor NF2) controlling a kinase cascade (MST1/2 and LATS1/2) that regulates the nuclear-cytoplasmic shuttling of YAP and TAZ. When signaling is shut down, YAP and/or TAZ are activated and released into the nucleus where they predominantly engage the TEAD family of transcription factors[3] and drive transcriptional activation from enhancer elements[4–6].

Besides their critical role in organ growth and tissue differentiation[7,8], YAP and TAZ are constitutively activated in many cancers, driving proliferative, pro-survival and invasive programs[9–11]. YAP/TAZ constitutive activation is driven by genetic aberrations in the

[1]Disease Area Oncology, Novartis Institutes for Biomedical Research, Basel, Switzerland. [2]Disease Area Oncology, Novartis Institutes for Biomedical Research, Cambridge, MA, USA. [3]Chemical Biology and Therapeutics, Novartis Institutes for Biomedical Research, Basel, Switzerland. [4]Chemical Biology and Ther-apeutics, Novartis Institutes for Biomedical Research, Cambridge, MA, USA. [5]Friedrich Miescher Institute for Biomedical Research, Basel, Switzerland. [6]Faculty of Sciences, University of Basel, Basel, Switzerland. [7]Present address: Alltrna Inc., One Kendall Square, Cambridge, MA, USA. [8]Present address: Helmholtz Pioneer Campus, Helmholtz Zentrum München GmbH German Research Center for Environmental Health, Neuherberg, Germany. [9]Present address: Roche Pharmaceutical Research and Early Development, Basel, Switzerland. ✉e-mail: giorgio.galli@novartis.com

Hippo pathway in a small number of cancer patients[12]. Most notably, up to 40% of Malignant Pleural Mesothelioma (MPM) cases bear mutations in upstream regulators of the Hippo Pathway, such as loss-of-function mutations in NF2 or LATS kinases[13]. Additionally, translocations involving YAP/TAZ have been reported in a rare soft tissue sarcoma called hemangioendothelioma[14,15].

Several alternative pathways have been suggested to cause activation of YAP/TAZ in a variety of cancers. These stimuli include mechanosensing, metabolism, cell adhesion or GPCRs[16]. Additionally, oncogenic mutations outside the Hippo pathway have been associated with YAP activation. A prototype example is Uveal Melanoma (UM), an ocular tumor involving aberrant proliferation of melanocytes. Uveal Melanoma is almost exclusively defined by activating mutations in the heterotrimeric G-protein alpha subunits *GNAQ* and *GNA11*[17,18]. These driver oncogenes activate Trio-Rho/Rac signaling to promote actin polymerization and ultimately activate nuclear YAP/TEAD transcription in a Hippo-independent manner[19].

Thereby, while both MPM and UM bear genetic alterations that promote YAP activation they do so by different pathways. If YAP engages different molecular mechanisms or genomic targets to exert its oncogenic functions in these different cancer settings remains a critical question to better understand the role of YAP in cancer.

Here we report that, while YAP is essential in both MPM and UM, its interaction with TEAD is critical only in MPM, suggesting that YAP can engage different mechanisms to drive oncogenesis in a disease displaying Hippo-independent YAP activation. We perform systematic functional characterization of YAP-bound regulatory elements (YREs) in both lineages and identify shared and lineage-specific sites involved in cancer-specific oncogenic programs. In MPM, we report critical YREs in loci controlling MAPK-responsive TFs, and enriched co-occupancy of such TFs and YAP at functionally relevant sites, translating in synergistic efficacy of combining TEAD and MAPK pathway inhibitors. Conversely, in UM we demonstrate that YREs rewire a neural-crest-derived network of melanocytic transcription factors (including MITF, SOX10 and PAX3) to promote a feedforward loop for cell proliferation. Additionally, we identify a set of lineage-specific YREs in loci of lineage-shared oncogenes such as *MYC* and *CCND1*. In conclusion, our data points to lineage-specific mechanisms engaged by YAP-driven regulatory elements. Considering the recent development of TEAD inhibitors[20], our study demonstrates that functional evaluation of the transcriptional regulatory networks engaged by transcriptional co-factors can support the design of therapeutic strategies in specific diseases.

## Results

### YAP engages different CREs in different cancer types

A variety of cancers display YAP nuclear activation and upregulation of YAP/TEAD targets. However, only MPM displays a significant fraction of patients with alterations in Hippo-signaling pathway[13]. We use uveal melanoma as a prototypical cancer lineage bearing a single oncogenic mutation ultimately driving Hippo-independent YAP activation. With these two model systems at hand (MPM and UM), we asked if YAP engages similar or divergent molecular mechanisms to drive its oncogenic program according to its activating signal. As previously reported[19,21], shRNA-mediated knockdown of YAP leads to prominent decrease in cell proliferation and concomitant downregulation of the canonical Hippo target gene *CYR61* in cellular models of both lineages (Figs. 1a and S1A). However, exposure of UM, MPM, non-small cell lung cancer (NSCLC), hepatocellular carcinoma (HCC), pancreatic ductal adenocarcinoma (PDAC) and skin melanoma cell lines to a TEAD inhibitor[22] demonstrated selective sensitivity of MPM cell lines compared to UM and other cells (Fig. 1b), suggesting additional mechanisms underlying YAP sensitivity in UM beyond TEAD engagement.

We characterized YAP/TAZ-driven cistrome in UM and MPM cells by mapping the genomic occupancy of YAP, TAZ and TEAD in two cellular models of each lineage. Genome-wide correlation revealed clustering of samples based on tissue origin with further clustering according to the specific cellular models (Fig. 1c). We additionally profiled our cellular models by ChIP-seq for histone modifications associated to enhancers, promoters or RNA-Polymerase II subunit RPB1. As expected, the signal of enhancer marks (H3K27ac and H3K4me1) was better at distinguishing cancer lineages compared to active promoter marks (H3K4me3 and total RNA Pol II) (Fig. S1B), in line with the previously reported role for enhancer marks in dictating cell type specificity[23]. We then analyzed the nature of YAP-positive peaks in the genome. YAP signal overall scaled with H3K27ac signal, confirming its predominant role as transcriptional co-activator and, as previously reported, the majority of YAP sites were enriched with the enhancer histone modification H3K4me1 and devoid of the promoter mark H3K4me3[6], independently of the cellular model considered (Figs. 1d and S1C). Differential binding analysis of YAP signal between UM and MPM models revealed a substantial fraction of YAP sites specific for either UM or MPM (Fig. 1e). While YAP occupies the loci of canonical target genes in both lineages, we observed YAP binding in the proximity of tissue-specific oncogenes/markers such as *AXL* and *SOX10* (Fig. 1e, f). In summary, our data demonstrate that, while YAP is necessary for proliferation of both MPM and UM cells, its proliferative functions might be dictated by the engagement of lineage-specific CREs.

### Functional interrogation of YAP-bound regulatory elements

We then sought to interrogate the functional role of YREs in MPM and UM cells. To do so, we built a consensus set of YAP-peak summits from the union of the YAP binding sites identified by ChIP-seq ($n > 135,000$). Next, we designed a comprehensive CRISPR library containing approximately 160,000 sgRNAs targeting the most significant YAP-peak summits among the four cell lines ($n_{total\ peaks} = 18039$, $-Log10FDR > 49$; 2941 peaks common to both lineages, 6389 mesothelioma-specific peaks, 2119 uveal melanoma-specific peaks, 2938 cell line-specific peaks and 3652 peaks in other distributions) (Fig. S2A–C and Supplementary Data 1). Our library design resulted in a median of 10 sgRNAs per YAP peak summit (Fig. S2D), distributed across each summit with a median distance of 61 bp between consecutive sgRNAs (Fig. S2E, F). We complemented our library with non-targeting (NT) guides and additional sgRNAs targeting the promoter of pan-lethal (PL) genes[24], Hippo-signaling components (HS)[25,26] and canonical YAP/TEAD target genes (Y/T)[27].

Using such library, which we called YAP-bound regions in Mesothelioma and Uveal Melanoma – CRISPR interference (YMCi), we infected the UM model 92.1 and the MPM model NCI-H2052, engineered to express dCas9-KRAB (CRISPRi), and harvested cells at day 8, 15 and 22 post-infection to evaluate sgRNA representation (Fig. 2a, b and Supplementary Data 2, 3). Independent analysis of the two library pools and biological replicates demonstrated excellent quality of the screens with high correlation of representation of common sgRNAs between the two pools and globally high correlation between biological duplicates (Fig. S3A, B). Importantly, for both cellular models, we observed negligible effects exerted by non-targeting sgRNAs, while pan-lethal sgRNAs displayed progressive loss of representation over time (Figs. 2b and S3A), further confirming the quality of the screen. Analysis of sgRNAs targeting Hippo-Pathway components or YAP/TEAD target genes confirmed the dependency of these cellular models on YAP, while TEAD dependency was restricted to NCI-H2052 cells (Fig. S4A), in line with the observed selective sensitivity to TEAD inhibitor (Fig. 1b). Only a subset of YAP/TEAD target genes belonging to cell cycle processes scored in both models, while the most prominent target genes and biomarkers for YAP activity (*CTGF*, *CYR61*, *AMOTL2*) seemed dispensable for in vitro cell proliferation (Fig. S4B).

Within experimental sgRNAs, as previously reported for Estrogen Receptor[28], only a small fraction of perturbations displayed significant scoring in UM and MPM, suggesting that only a subset of YREs is necessary for cell proliferation (Fig. 2b).

We then compared the dropout of each sgRNA between the two lineages and observed common and specific dependencies between the two cell lines (Fig. S4C). To identify specific dependencies, we performed differential analysis. Comparison of the YMCi screens at each timepoint yielded a small number of sgRNAs differentially enriched in NCI-H2052 and 92.1 (NCI-H2052

$n_{(d22)} = 311$; 92.1 n $n_{(d22)} = 1262$, Abs(log2FC) > 1, FDR < 0.05) (Figs. 2c and S4D). For downstream analyses and validation, we selected a set of sgRNAs consisting of the top 4 sgRNAs targeting the YAP-peaks identified as commonly or differentially scoring in the primary screen (Fig. S4C, E and Supplementary Data 3) and several controls (see next paragraph for experimental validation). Differential-lineage transcription-factor motif analysis of these scoring regions uncovered a trend for enrichment of motifs recognized by MAPK-responsive TFs in MPM, in line with previously reported genomic proximity between TEAD and AP-1

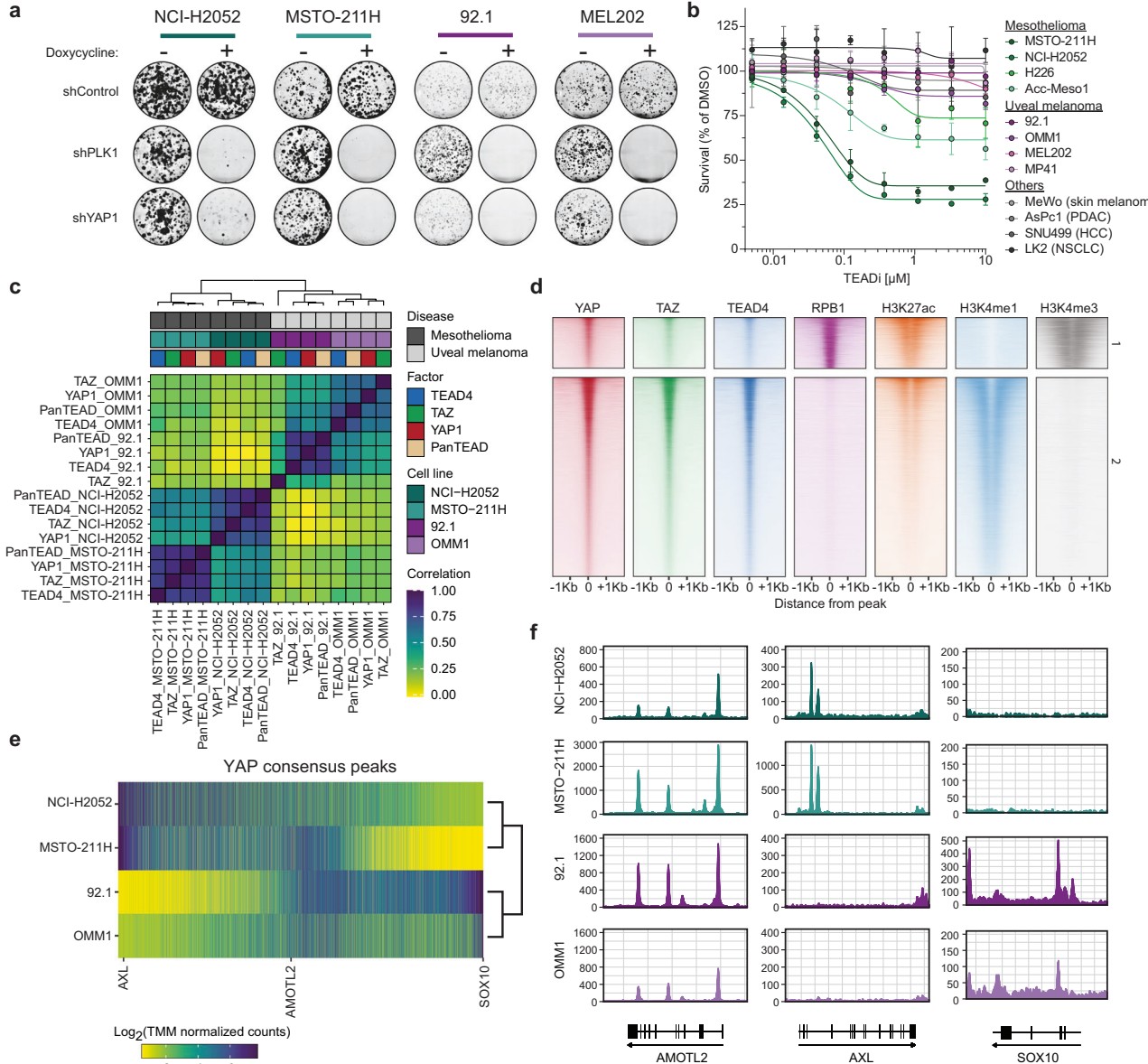

**Fig. 1 | Mesothelioma and uveal melanoma cells lines show distinct YAP1 occupancy patterns. a** Colony formation assays of mesothelioma (NCI-H2052 and MSTO-211H) and uveal melanoma (92.1 and MEL202) cell lines expressing a doxycycline-inducible shRNA against a neutral control (shControl), an essential gene (shPLK1) and YAP1 (shYAP1). Cells were incubated for 12 days in the presence or absence of doxycycline. **b** Proliferation assay of mesothelioma (shades of green), uveal melanoma (shades of purple) and cell lines from other lineages (shades of gray) treated with increasing doses of a TEAD inhibitor for 5 days. Y-axis represent % of cells surviving compared to DMSO treatment. Error bars represent standard deviation of 3 independent replicates. Source data are provided as a Source Data file. **c** Heatmap depicting genome-wide correlation of TMM normalized read counts

of YAP1, TAZ, TEAD4 and PanTEAD ChIP-seq in mesothelioma and uveal melanoma cell lines. **d** Heatmaps representing ChIP-Seq signal of YAP1, TAZ, TEAD4, RPB1, H3K27ac, H3K4me1, and H3K4me3 around YAP1 ChIP-Seq peak summits in MSTO-211H cells. The peak summits have been clustered by their H3K4me3, H3K4me1, and Rpb1 signal (1 = promoter regions, 2 = enhancer regions). The summits have been arranged in decreasing order of YAP1 signal within each cluster. **e** Heatmap depicting YAP1 peak occupancy in mesothelioma and uveal melanoma cell lines at the consensus YAP1 peaks (union of YAP1 peaks from all samples). Peaks were ranked horizontally based on the Δ(Mesothelioma/Uveal) signal. **f** YAP1 occupancy at the AMOTL2, AXL and SOX10 genes in mesothelioma and uveal melanoma cell lines.

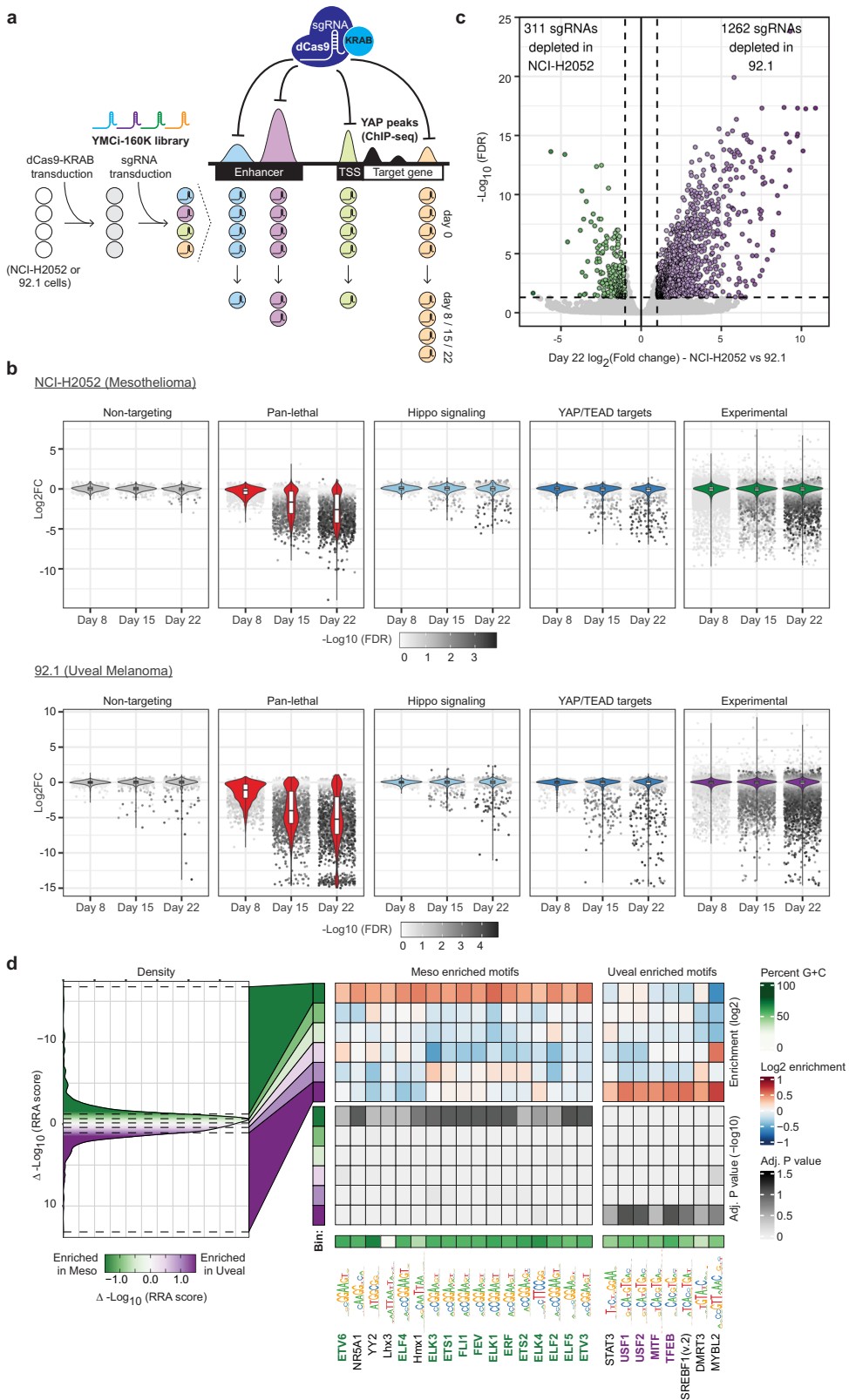

sites[4,5] (Fig. 2d and Supplementary Data 4). In sites specifically scoring in UM, we observed a trend for enrichment of the CAYGTG motif (Fig. 2d). Of note, MITF, the master regulator of melanoma, is among TFs recognizing such motifs. In summary, large-scale functional interrogation of YREs in two different cancer lineages uncovers common and differential functional relevance of YAP-controlled regulatory elements.

### Identification of lineage-specific functional YREs

To validate and extend the identification of lineage-specific YREs, we generated a validation CRISPRi library (n = 13000 sgRNAs) based on selected hits mentioned above and screened four cell lines (two for each lineage) over time, following same strategy used for our YMCi-160K library (Supplementary Data 5 and 6). Across all models, we observed progressive depletion of a subset of sgRNAs (Fig. S5A) with

**Fig. 2 | CRISPRi screens for evaluating the essentiality of YAP1-bound regions in mesothelioma and uveal melanoma cells. a** Schematic representation of the YMCi screen workflow. **b** Enrichment (Log2-fold change) of sgRNAs within the various control and experimental pools ($n_{NT} = 496$, $n_{PL} = 1030$, $n_{HS} = 419$, $n_{YT} = 605$, $n_{Experimental} \approx 160,000$), at day 8, day 15 and day 22 timepoints compared to day 0, in both mesothelioma (NCI-H2052) and uveal melanoma (92.1) screens. Dots represent scoring for individual sgRNAs. Significance of depletion is represented as -Log10FDR in gray colored scale. Boxplots represent median and first and third quartiles, and whiskers extend to 95th percentile. **c** Volcano plot showing the comparison of sgRNA depletion between H2052 (MPM) and 92.1 (UM) cells.

Significantly depleted sgRNAs (FDR < 0.05) are colored (NCI-H2052; green dots) (92.1; purple dots). Gray dots represent non-differentially represented sgRNAs. X-axis represents Log2(Fold change) between the two cell lines. Y-axis depicts the log10-transformed FDR value. **d** Transcription-factor motif enrichment analyses for YREs scoring specifically in the mesothelioma or uveal melanoma screens. On the left side, binned density of the RRA score difference between mesothelioma and uveal melanoma scoring regions, with equal size bins of 2000 sites. On the right side, enrichment and significance (T-test corrected by BH) heatmaps of motifs (columns) across bins (rows).

---

overall substantial correlation between primary (YMCi-160K library) and secondary screen (YMCi-13K library) (Fig. S5B). Importantly, correlation analyses of the secondary screen reveal lineage-specific clustering (Fig. S5C), further validating our strategy to identify lineage-specific functional YREs. Additionally, we confirmed the dependency of all models to depletion of YAP, while the dependency to different TEAD homologs was restricted to mesothelioma models (Fig. S5D). We then called YAP peak-level hits displaying decreased fitness score over time for each model (Fig. 3a and Supplementary Data 7) and intersected the data to establish a list of YREs commonly scoring in both lineages (scoring in 3 or more cell lines) or scoring specifically in MPM or UM. We noticed that the list of common hits included mostly regions close to the TSS of cell cycle- or apoptosis-related genes, while lineage-specific hits comprised a larger portion of putative distal regulatory elements (Fig. 3b, c; Supplementary Data 8).

To validate the results of our screen, we set up a time-resolved FACS-based competition assay (Fig. S6A). While a non-targeting sgRNA (sgNTC) was stably maintained in the cell population, sgRNAs targeting the promoter of RPL14 or YAP itself strongly dropped out over time, with concomitant impact on expression of YAP target genes (Fig. S6B and S6C). With this system, we confirmed that knockdown at the TSS of *NDC1* and *AAAS*, two components of the nuclear pore complex, or silencing a putative enhancer of the nuclear receptor *NR2F2*, decreased the proliferation of mesothelioma cell lines but not of uveal melanoma (Fig. S6D, F). Conversely, uveal melanoma cells were more sensitive to the knockdown of *NR4A3* or *KEAP1* (Fig. S6E, F), mirroring the results of our screens. Further analysis of genomic locations and assignment of scoring YREs to TSSs revealed other interesting hits. Most evidently, several YREs belonging to the *MYC* locus (numbered asterisks in Figs. 3c and 4a), within 2 Mb distance from *MYC* TSS, scored in both lineages. Importantly, we observed lineage-specific scoring YREs concomitant with lineage-specific YAP occupancy. The most prominent hits were a mesothelioma-specific YRE 440 Kb downstream of *MYC* TSS (Fig. 4a, b, green highlighted region *3-5) and a uveal melanoma-specific region 1.8 Mb downstream (Fig. 4a, c, purple highlighted region *9). The lineage-specific essentiality of these regions was already apparent in the primary YMCi-160K screen, where several sgRNAs (>4) were targeting these YREs (Fig. S7A). Using sgRNA competition assays we confirmed that silencing the 440 kb and 1.8 Mb enhancers affected the viability of mesothelioma and uveal melanoma cells, respectively, with a corresponding lineage-specific impact on *MYC* expression (Fig. S7B, C). Importantly, high-resolution HiC maps and H3K27ac HiChIP in both lineages revealed specific loops between MYC TSS and the lineage-specific YREs bearing prototypical enhancer features as assayed by ChIP-seq and Cut&Tag (Fig. 4b, c). Our data demonstrate that, while YAP is expressed and functionally relevant in both lineages, it can engage distinct regulatory elements converging onto a broadly active oncogene.

### YAP/TEAD cooperates with MAPK-responsive TFs in Mesothelioma
Among the Mesothelioma-specific hits we observed YREs in the loci of three MAPK-responsive TFs: *JUN*, *FOSL1* and *FOSB* (Figs. 3c, 5a and S8A). In the *JUN* locus, we found one site upstream of its TSS (−361 Kb)

highly enriched for H3K27ac and H3K4me1 histone enhancer marks and displaying mesothelioma-specific YAP occupancy, as well as binding by the AP-1 TFs FOSL1 and JUN itself (Fig. 5a). The identified YRE was also engaged in enhancer-promoter looping, as verified by HiC and H3K27ac-HiChIP. Similar observations could be made for the *FOSL1* and *FOSB* loci (Fig. S8). Targeting these YREs with single sgRNAs confirmed a decrease in cell growth of mesothelioma cells and a concomitant downregulation of their predicted target genes' expression (Figs. S9A and S9B, C). Additionally, the signal of H3K27ac, YAP1, JUN and FOSL1 was enriched in YREs specifically scoring in mesothelioma cells and not in common- or UM-specific hits (Fig. 5b). Genome-wide analysis of Cut&Tag for multiple TFs revealed high correlation of YAP signal with JUN and FOSL1 in MPM cells (Fig. S9F). These data suggest that these MPM-specific YREs serve as transcription-factor binding-hubs with essential regulatory roles in mesothelioma cells. The identification of YREs connected to MAPK-responsive TFs further suggested an intimately connected network between Hippo and MAPK signaling in mesothelioma cells. Indeed, TCGA analyses showed increased expression of several MAPK TFs in mesothelioma, specifically the ones belonging to the ELK, ELF and AP-1 subfamilies (Fig. S9G) and we observed synergistic anti-proliferative effects with combined treatment of a TEAD inhibitor and several agents inhibiting MAPK pathway at different levels (Fig. 5c). In summary, our functional epigenomics screen reveal an additional point of interaction between YAP and MAPK pathway specifically in mesothelioma cells, further prompting the exploration of combinatorial treatments in this disease.

### YREs rewire a network of uveal melanoma-specific master regulators
Uveal melanoma-specific hits were enriched for YREs present in the loci of three master regulators of gene expression in the melanocytic lineage and melanoma: PAX3, SOX10 and MITF (Fig. 3c). In the SOX10 locus we identified an enhancer within the intronic region of POLR2F and −54 Kb away from the SOX10 TSS (Fig. 6a). Importantly, we demonstrate that targeting this site using CRISPRi leads to specific downregulation of *SOX10* without altering the expression of the pan-essential gene *POLR2F* (Fig. S10A, B). Notably, this site not only controls *SOX10* expression, but it also displays co-occupancy of SOX10 itself, together with YAP and the other melanocytic transcription factors PAX3, MITF and TFAP2A. High-resolution HiC and H3K27ac HiChIP confirmed that the *SOX10* −54Kb site loops specifically to the promoter of *SOX10* within a CTCF-delimited locus (Fig. 6a). In the loci of *PAX3* and *MITF*, we also found several YREs with: (1) UM-restricted YAP binding, (2) organization in 3D-proximity to the promoter of the corresponding genes and (3) co-occupancy of the key melanocytic TFs (Fig. S10C). Silencing these enhancer regions (*PAX3* +138 Kb; *MITF* −115 Kb and −97 Kb) using single sgRNAs confirmed the UM-specific impact on cell proliferation (Fig. S10D and S10E) and resulted in downregulation of *PAX3* and *MITF* endogenous genes (Fig. S10F and S10G). These data suggest that a coordinated network of self- and cross-regulated melanocytic transcription factors engage YAP in uveal melanoma cells to sustain its proliferative program. In agreement with this observation and analogous to the situation in mesothelioma, YREs specifically scoring in uveal melanoma cell lines were enriched for

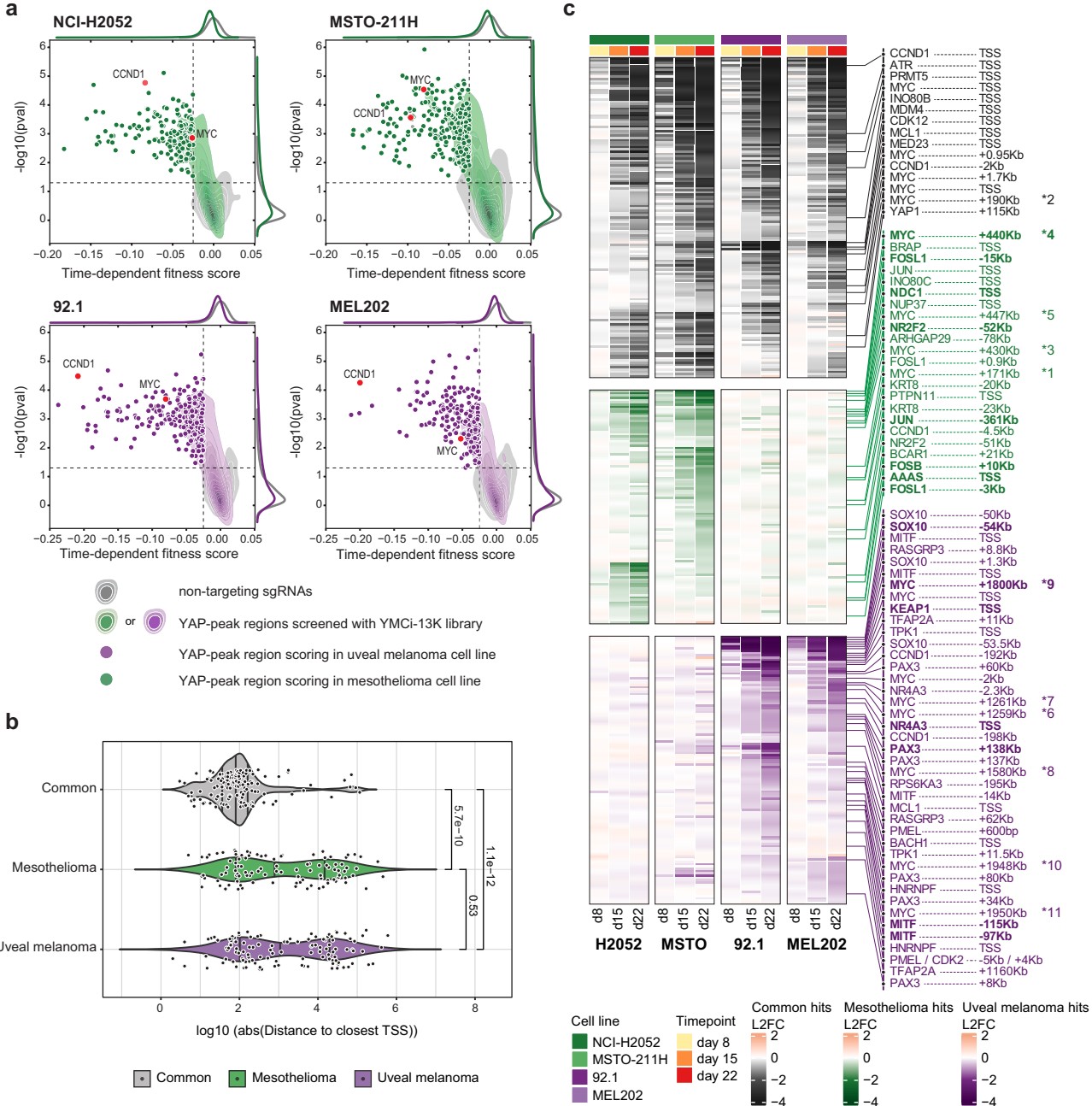

**Fig. 3 | Secondary CRISPRi screens identify common essential and lineage-specific vulnerabilities. a** Contour-dot plots showing the scoring of YAP-bound regions in secondary screens of mesothelioma (NCI-H2052 and MSTO-211H) and uveal melanoma (92.1 and MEL202) cell lines. X-axis represents the metric for growth rate and Y-axis depicts the log10-transformed *p* value. For two exemplary regions, dots are colored in red and labeled according to the closest TSS (CCND1 and MYC). **b** Violin plot depicting distance to closest TSS of regions scoring in all cell lines (common = gray), MPM-specific (green) and UM-specific (purple).

Distance to TSS is shown as Log10(bp) of absolute distance. Vertical lines represent median, first and third quartiles. *P* values are based on two-sided Wilcoxon's rank-sum test. **c** Heatmap showing LogFC depletion of YREs commonly scoring in both lineages (black color scale) or regions specific to mesothelioma (green color scale) or uveal melanoma (purple color scale). On the right hand-side, selected hits regions are labeled according to the distance to closest annotated TSS or TSS of putative target, informed by inspection of HiC/HiChIP loops. Regions marked with an asterisk belong to the MYC locus and can be visualized in Fig. 4.

H3K27ac, YAP, PAX3, SOX10 and MITF Cut&Tag signal (Fig. 6a, b). Importantly, analyses of TCGA data confirmed that high expression of *PAX3*, *SOX10*, *MITF* and *TFAP2A* are defining features of uveal melanoma patient samples (Fig. S11A) and such TFs have been validated as selective cancer dependencies in UM cell lines by large-scale functional genomics screens (Fig. S11B).

In summary, our data suggests that YAP broad oncogenic functions might be attributed to engagement of lineage-specific YREs controlling the expression of common oncogenes like MYC or CCND1,

thus defining shared dependencies. Additionally, our findings reveal the integration of YAP into lineage-specific regulatory networks, providing rationale for designing therapeutic modalities aimed at targeting YAP signaling, beyond its interaction with TEAD (Fig. 6c).

## Discussion

The Hippo pathway is a signaling cascade that converges on the transcriptional co-activators YAP and TAZ engaging TEAD transcription factors in the nucleus. YAP has been shown to be activated by

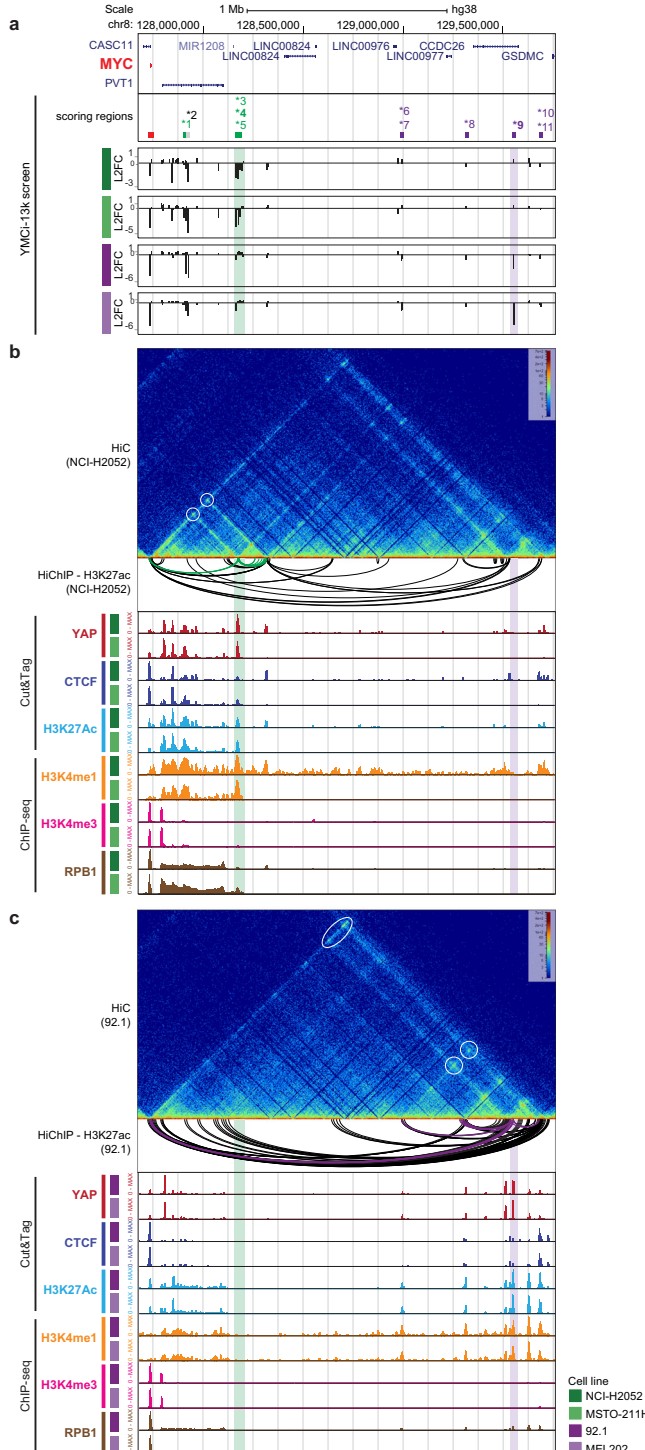

**Fig. 4 | Secondary-screen reveal functional YREs in the MYC locus. a** Genome browser snapshot of the MYC locus, with aligned bigwig tracks representing scoring regions and Log2-fold change of sgRNAs depletion (250 bp span for each sgRNA) at day 22 for the YMCi-13K screen in four cell lines. **b, c** HiC interaction frequencies heatmap, HiChIP-H3K27ac significant loops (FDR < 0.05), Cut&Tag tracks for YAP1, CTCF and H3K27ac, and ChIP-seq tracks for H3K4me1, H3K4me3 and the RNA polymerase II subunit RPB1 in MPM (**b**) and UM (**c**) cells. Lineage-specific regions and HiChIP-H3K27ac loops are colored in green for MPM (NCI-H2052) or purple for UM (92.1).

Hippo-dependent and -independent mechanisms. Here we show that the interaction between YAP and TEAD is essential in a Hippo-dependent cancer but dispensable in a prototypical Hippo-independent cancer, such as uveal melanoma. This specificity can be

accounted for by the differential transcriptional networks we identify to be engaged by YAP-regulatory elements in these two disease models, revealing a connection between MAPK and YAP at the genomic level in mesothelioma, and the rewiring of a network of uveal melanoma-specific master regulators at YREs in uveal melanoma. Our findings highlight the value of systematic functional interrogation of regulatory elements controlled by a broadly expressed transcriptional co-factor.

The Hippo pathway has been reported as a major regulator of mammalian organ size and adult stem cells biology[7, 8]. While being critical for proper tissue development and differentiation, this pathway has been of great interest from a therapeutic perspective, particularly in regenerative biology, fibrosis and oncology. In cancer biology, the Hippo signaling has been shown to be genetically deregulated in a small subset of cancers such as mesothelioma with *NF2* or *LATS* kinases mutations[13] or hemangioendotheliomas characterized by translocations involving YAP or TAZ[14,15]. Despite these rare genetic aberrations, a wide variety of cancers display high activity of the downstream effectors YAP/TAZ[12], suggesting a strong potential for therapeutic intervention. For this reason, several molecules have been developed targeting the transcriptional effectors of the pathway TEADs[20]. While these molecules are moving steadily towards the clinic in MPM and other *NF2* mutant cancers[29,30], we demonstrate that their applicability to other YAP/TAZ-driven cancers might be limited and requires further evaluation.

In MPM we validate the critical role of YAP-TEAD interaction both by genetic means (using CRISPRi at the promoter of TEAD genes) and by chemical tools (TEAD lipid-binding pocket inhibitors). Our genome-scale analyses identify an integrated feedforward loop, stemming from YAP-bound regulatory elements, controlling the expression of MAPK-responsive transcription factors. Indeed, at MPM-specific YREs we observe enrichment of occupancy of the same MAPK TFs, indicating an intricate regulatory network between YAP/TAZ-TEAD and MAPK, extending far beyond the mere co-occurrence of AP-1 motifs flanking TEAD motifs in the genome[4,5] or p38-dependent TEAD translocation[31]. Additionally, while AP-1 TFs induction has been previously reported in UM[32], we observe exquisite lineage specificity for YREs connected with MAPK TFs in MPM and suggest that the previously reported sensitivity of UM cells to MAPK inhibitors might be Hippo-signaling independent. Our findings further validate and extend the importance of the formerly suggested therapeutic combinations between TEAD inhibitors and MAPK signaling inhibitors[33–36], specifically in MPM.

Uveal melanoma is another cancer dependent on YAP[19,37], whilst displaying a Hippo-independent mechanism of YAP activation. We validate that YAP activity is critical for survival of UM cells as well as for transcription of canonical target genes, as previously reported[19,37]. YAP canonical target genes (e.g. *CTGF*, *CYR61*, *AMOTL2*) are well known to be regulated by the interaction between YAP and TEAD[3]. While these canonical target genes are an excellent proxy for Hippo-dependent pathway activity, we show that they are not able to explain the lineage-restricted sensitivity to TEAD inhibitors. We report that in UM, an entire cluster of transcription factors regulating neural-crest and melanocytic differentiation[38] engages YAP responsive elements. Thereby, while TEADs are ubiquitously expressed and their occupancy tends to correlate with that of YAP genome-wide[3], it is conceivable that other complexes can engage YAP on chromatin to induce lineage-specific transcriptional activation.

In both MPM and UM, we identified regulatory elements within the *MYC* locus which were also engaged by YAP in a lineage-specific manner and involved in long-range transcriptional regulation of *MYC*. Numerous tissue-specific enhancers of *MYC* have been described that can lead to MYC hyperactivation[39]. Here we show that the MYC-LASE enhancer, previously described in lung cancer, and MYC-BENC, reported in Acute Myeloid Leukemia (AML), are also functional in MPM and UM, respectively. Lineage-specific targeting of MYC in

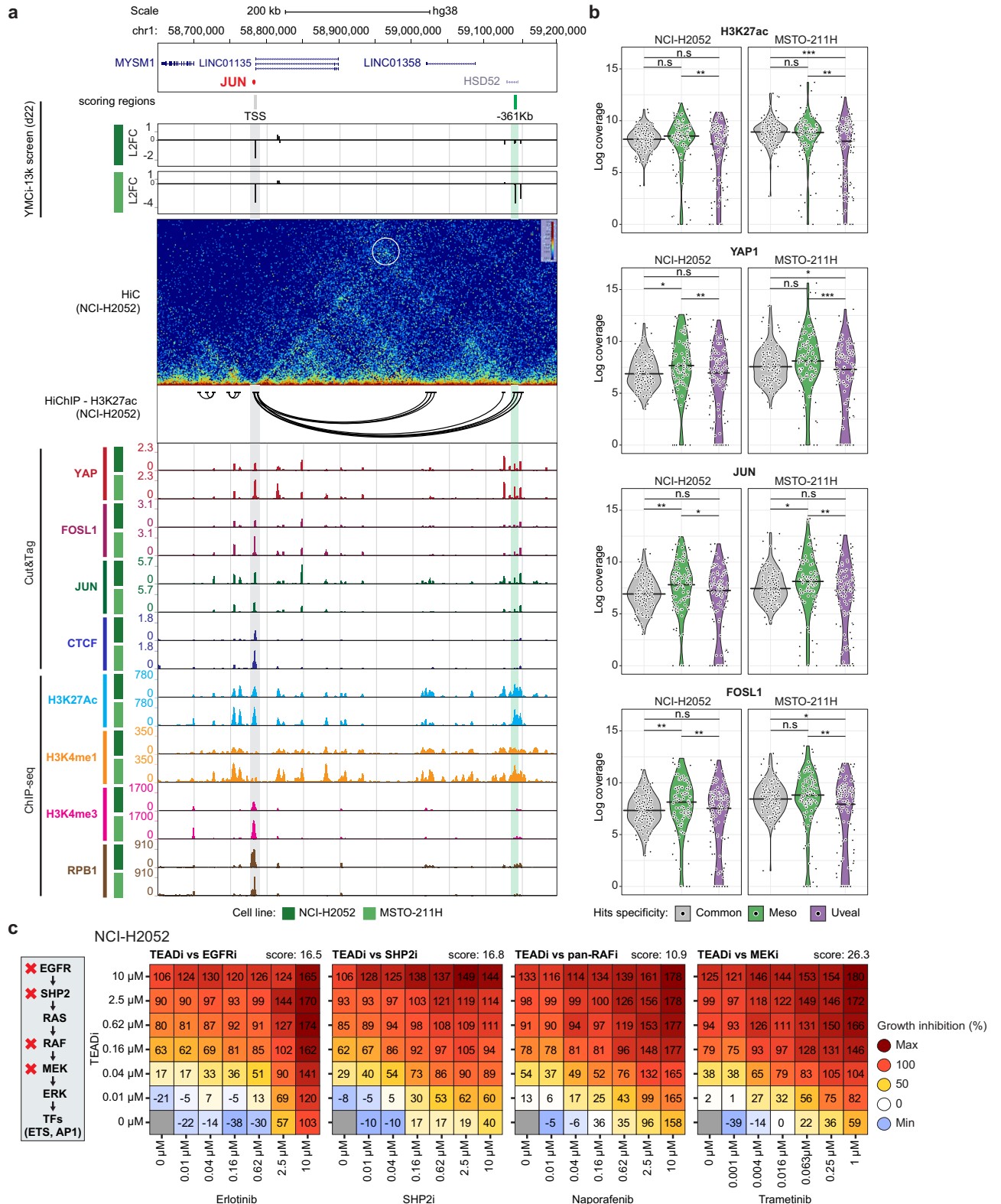

these diseases could potentially be achieved by taking advantage of these enhancer regions, as is currently being pursued, for example, in sickle cell disease by targeting an erythroid-specific enhancer of *BCL11A*[40,41].

Functional interrogation of regulatory elements is becoming prominent thanks to the advances in molecular biology and CRISPR technology[42]. Indeed, several CRISPR-based screens have revealed enhancers regulating specific genes, or transcriptional networks

engaged by lineage-specific transcription factors[28,43–45]. However, one of the big questions in the field is how ubiquitously expressed transcriptional co-factors engage specific TFs or enhancers. Recent work highlights that different regulatory elements and co-factors can engage each other in an unspecific and specific manner[46–48]. Although a functional epigenomic screen on YREs has been performed in non-transformed MCF10A cells[49], here we provide evidence of how functional epigenomics can reveal lineage-specific transcriptional

**Fig. 5 | YAP engages YREs in loci of MAPK genes to reinforce signaling.**
**a** Genome browser snapshot of the JUN locus, showing YMCi-13K screen results (scoring regions and Log2-fold change at day 22, represented with bigwig tracks with 250 bp span), HiC data, HiChIP-H3K27ac significant loops (FDR < 0.05), Cut&Tag tracks for YAP1, FOSL1, JUN and CTCF, and ChIP-seq tracks for H3K27ac, H3K4me1, H3K4me3 and the RNA polymerase II subunit RPB1. Scoring regions validated in single assays are color-shaded in green. **b** Cut&Tag signal coverage of indicated marks/factors (H3K27ac, YAP1, FOSL1 or JUN) in the group of regions that commonly score in all cell lines (gray color), or that score specifically in mesothelioma (green color) or uveal melanoma (purple color). Horizontal line in violin plot represents median of data distribution. Significance was assessed by two-sided Wilcoxon test (*$P < 0.05$; **$P < 0.01$; ***$P < 0.001$, and not significant (n.s.) $P > 0.05$). Source data are provided as a Source Data file. **c** Combination data heatmaps for treatment of NCI-H2052 mesothelioma cells with a TEAD inhibitor and several MAPK pathway inhibitors. Growth inhibition (%) is represented in a color scale, in which 100% corresponds to proliferation arrest and maximum value to cell death. Loewe synergy scores are shown on top right corner of each heatmap.

networks controlled by an oncogenic transcriptional co-factor activated by different genetic aberrations. While we provide extensive orthogonal evidence on how our screens unveil principles of YAP-dependent regulation, our work largely relies on the use of the dCas9-KRAB system. Additional studies using advanced genetic engineering, or YAP-TEAD interface disruptor molecules will be needed to corroborate the direct contribution of YAP in the regulation of YREs and downstream genes. Furthermore, future work will be needed to characterize the upstream mechanisms dictating YAP distribution in the genome, its preferential association to different complexes and the structure-function relationship within such YAP-containing enhanceosomes.

# Methods
## Cell Culture
The mesothelioma cell lines (NCI-H2052 [ATCCCRL5915], MSTO-211H [ATCCCRL:2081], H226 [ATCCCRL:5826] and Acc-Meso1 [RikenRCB2292]), uveal melanoma cell lines (92.1 [Leiden University Medical Center - M.J. JagerCVCL_8607], MEL202 [ATCCCRL:3296] and OMM1 [Leiden University Medical Center - M.J. JagerCVCL_6939]) and the LK-2 [RikenRCB1970], SNU-449 [ATCCCRL:2234] and AsPc1 [ATCCCRL:1682] cell lines were cultured in RPMI-1640 medium supplemented with 10% fetal bovine serum (FBS), 2 mM L-glutamine, 1 mM sodium pyruvate, 0.1 mM of each non-essential aminoacid (NEAA) and 10 mM HEPES. MeWo cells were cultured in EMEM medium supplemented with 10% fetal bovine serum (FBS), 2 mM L-glutamine, 1 mM sodium pyruvate, 0.1 mM of each non-essential aminoacid (NEAA) and 10 mM HEPES. 293FT cells (ThermoFisher) were cultivated in DMEM supplemented with 10% fetal bovine serum (FBS), 2 mM L-glutamine, 1 mM sodium pyruvate, 10 mM HEPES. All culture media and supplements were from BioConcept.

## Compound testing
For TEAD inhibitor testing, cell lines were seeded in 96-well plates (Corning, 3903), 1500 cells per well, 24 h before treatment. For compound combination assays, NCI-H2052 cells were seeded in 384-well plates (Corning, 3570), 250 cells per well, 24 h before treatment. Test compounds (IK930[50], VT-104[51], Erlotinib, Naporafenib, Trametinib and SHP2i) were distributed in a randomized manner into the assay plates using the HP D300 Digital dispenser (Hewlett-Packard), in an 8-point threefold or 6-point fourfold serial dilution starting at top concentration of 1uM for Trametinib or 10 µM for all other compounds, each concentration tested in triplicate. No statistical method was used to predetermine sample size and no data were excluded from the analyses. DMSO was used as control and DMSO content was normalized to highest volume in all compound treated wells. After an incubation period of 3 days at regular cell culture conditions (37 °C, 5% CO₂), cell viability was assessed using CellTiter-Glo 2.0 assay (Promega) according to manufacturer's protocol. Luminescent signal was recorded with an Infinite M200 Pro instrument (Tecan). Background luminescent signal of cells before treatment was also recorded, following strategy described above, and values used for normalization. The IC50 values were calculated using dose response curves with GraphPad Prism (GraphPad Software, LLC). Compound combination activity was determined based on Loewe dose additivity using a weighted synergy score (SS) calculation.

## Virus production and cell infection
Viral particles were produced by transfection of semi-confluent 293FT cells (ThermoFisher) with transfer plasmid and Lentiviral Packaging plasmid mix (Cellecta), using the TransIT-LT1 (MirusBio) transfection reagent and following manufacturer's protocol. Viral supernatant was harvested at 48 h and 72 h post transfection and target cells were infected using virus dilutions in the presence of polybrene (8 µg/ml).

## Cell engineering
Cells expressing doxycycline-inducible shRNAs were generated by lentiviral transduction of a modified pLKO-TET-ON plasmid, followed by puromycin selection (1 µg/ml). shRNA and sgRNA sequences used for single validation studies are listed in Supplementary Data 9. To generate CRISPRi competent cell lines stably expressing dCas9-KRAB, cells were transduced with a lentivirus delivering the vector pCMV-dCas9-KRAB-IRES-BlastR, followed by blasticidin (InvivoGen) selection (10 µg/ml).

## Cell competition assays
For competitive proliferation assays, CRISPRi cells were transduced with a lentiviral vector modified from the pLKO plasmid, allowing the expression of sgRNA and the RFP fluorescent marker (pNGx-U6-sgRNA-modifiedTracer-cPPT-UBC-RFP-T2A-Puromycin). The fraction of RFP-expressing cells was determined 3 days post-infection using flow cytometry, this value used as starting reference, and RFP population was monitored over time. Data was acquired in a Cytoflex S instrument (Beckman Coulter) and analyzed using the CytExpert software (Beckman Coulter, v2.4.028). The sgRNA sequences used in this study are provided in Supplementary Data 9. No statistical method was used to predetermine sample size, no data were excluded from the analyses and the Investigators were not blinded to allocation during experiments and outcome assessment.

## Colony formation assays (CFA)
Doxycycline-inducible shRNA-expressing cells were seeded at low density (2500 cells/well in 6-well plates) without doxycycline or with continuous treatment of doxycycline (100 ng/ml) for 12 days, after which cells were fixed with 4% formaldehyde for 10 min and colonies stained with crystal violet. The shRNA sequences used in this study are provided in Supplementary Data 9.

## Design of YMCi-160K and YMCi-13K libraries
Overall, the design of the sgRNA library targeting YAP sites followed multiple steps. First, we selected the most robust YAP ChIP-seq peaks for each cell line and created a consensus peakset. Next, we identified sgRNAs targeting such peaks with defined criteria. Finally, due to constrains in the number of sgRNAs that could be incorporated in libraries, we further restricted to guide RNAs which targeted the peak summits with the highest significance ($n_{sgRNAs} = 159612$; $n_{total peaks} = 18039$).

In detail, the YMCi-160K library was designed as outlined in Fig. S2A. YAP peaks from ChIP-seq were called using MACS2[52] peak calling in four cell lines (92.1, OMM1, NCI-H2052 and MSTO-211H). Due to variability in ChIP target concentration in each cell line and ChIP

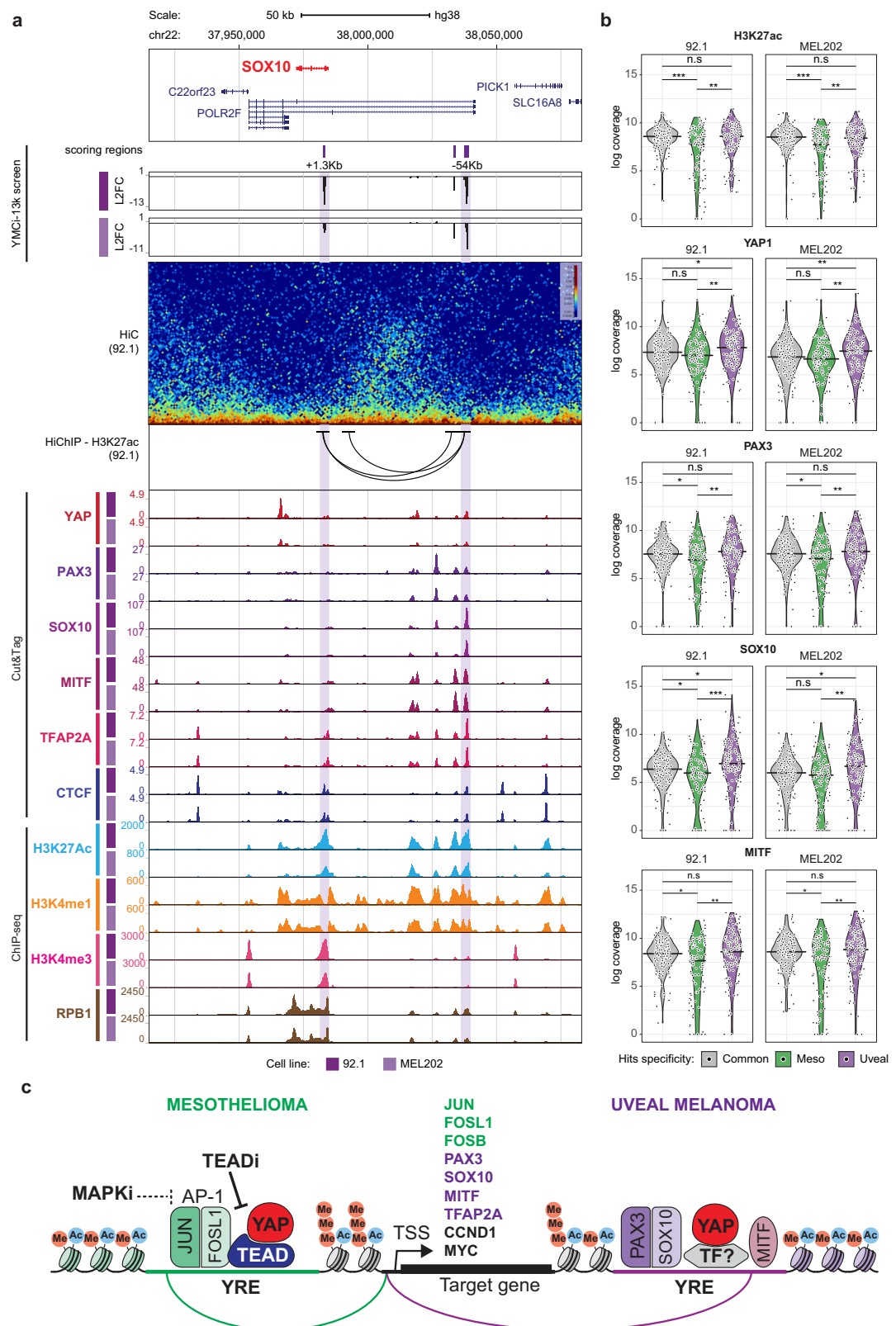

efficiency, optimal FDR cutoff for peak calling was obtained for each cell line using the following data-driven approach: (1) we plotted the distribution of FDR values as an empirical cumulative distribution of all peaks for each cell line, (2) we next fitted a curve (function smoothed.spline (nknots = 20, df = 15)), we identify the inflection point through the second derivative and retained all peaks above that threshold.

Peak union was obtained by pairwise overlaps with PlyRanges[53] annotating unique IDs for YAP-peaks (Union_ID) and coordinates for each peak summit (Absolute_Summit). All available sgRNA sequences bearing spCas9 PAM NGG were annotated using GuideScan[54] filtering for sgRNAs bearing no off targets allowing 1 mismatch. Due to the wide peak size, we narrowed our search on regions of ±300 bp of distance from YAP summits within each peak. Additionally, sgRNAs were

**Fig. 6 | YAP binds YREs in loci of melanocytic transcription factors in Uveal Melanoma. a** Genome browser snapshot of the SOX10 locus, showing YMCi-13K screen results (scoring regions and Log2-fold change at day 22 using bigwig tracks with 250 bp span), HiC data, HiChIP-H3K27ac significant loops (FDR < 0.05), Cut&Tag tracks for YAP1, PAX3, SOX10, MITF, TFAP2A and CTCF, and ChIP-seq tracks for H3K27ac, H3K4me1, H3K4me3 and the RNA polymerase II subunit RPB1. Scoring regions validated in single assays are color-shaded in purple. **b** Violin plots showing Cut&Tag signal coverage of indicated marks/factors (H3K27ac, YAP1,

PAX3, SOX10 or MITF) in the group of regions that commonly score in all cell lines (gray color), or that score specifically in mesothelioma (green color) or uveal melanoma (purple color). Horizontal line in violin plot represents median of data distribution. Significance was assessed by two-sided Wilcoxon test (*$P < 0.05$; **$P < 0.01$; ***$P < 0.001$, and not significant (n.s) $P > 0.05$). Source data are provided as a Source Data file. **c** Model depicting the findings of the study. YAP engages lineage-specific YREs to establish feedforward loops with specific transcription factors critical for lineage-specific cell proliferation.

filtered according to the following criteria: (1) absence of BbsI restriction enzyme site and (2) distance to previous sgRNA > 20 bp. Interpolation of the results of this filtering strategy with the entire dataset of summits resulted in the selection of two cutoffs for peaks targeted by sgRNAs: (1) FDR < $10^{-92}$ and (2) FDR > $10^{-92}$ and FDR < $10^{-49}$, in order to generate two pools containing each around 80,000 sgRNAs. We complemented our library with a control pool composed of non-targeting (NT) guides ($n = 496$) and additional sgRNAs targeting the promoter of pan-lethal genes (PL $n = 1030$; gene list according to common essential list[24]), Hippo pathway components (HS $n = 419$; gene list according to C5:GOBP_HIPPO_SIGNALING, May 2020)[25,26] and canonical YAP/TEAD target genes (YT $n = 605$; gene list according to CORDENONSI_YAP_CONSERVED_SIGNATURE)[27]. All sgRNA sequences for these sites were retrieved from hCRISPRi v2.1 library[55]. See Supplementary Data 1 for complete list of YMCi-160K library features.

The YMCi-13K library was designed to obtain a smaller library with an even number of sgRNAs per peak summit ($n = 4$) based on the top scoring sites from YMCi-160K screen. For the design, we used data from timepoint day 22 of the YMCi-160K CRISPRi screens in NCI-H2052 and 92.1 cells (Supplementary Data 3). We first subset depleted sgRNAs with p-val <0.1. Union_ID hits were defined based on at least 2 depleting sgRNAs (lineage-specific hits) or 3 depleting sgRNAs (common hits) with at least one of them being significant ($p < 0.05$). For all the hits, the 4 top scoring sgRNAs were selected to build YMCi-13K. The library was additionally supplemented with sgRNAs targeting Hippo-pathway (HS) or YAP/TEAD targets (YT) (5 sgRNAs per gene, 490 sgRNAs total), Pan-lethal (PL) genes (5 sgRNAs per gene, 480 sgRNAs total) and non-targeting sgRNAs (346 sgRNAs total), resulting in a library of 13000 sgRNAs. See Supplementary Data 5 for complete list of YMCi-13K library features.

## Construction of YMCi-160K and YMCi-13K libraries
The YMCi-160K ($n_{(pool\ 1)} = 82,879$ and $n_{(pool\ 2)} = 81,833$) and the YMCi-13K ($n = 13,000$) libraries were purchased from Twist Bioscience. The designed 60-bp single-stranded DNA oligos contained a 20-bp sgRNA flanked by sequences with BbsI recognition sites (5′-GCCATCCAGAA-GACTTACCG-3′ and 5′-GTTTCCGTCTTCACGACTGC-3′). The oligo pools were amplified using the NEBNext High-Fidelity 2× PCR Master Mix (New England BioLabs), in 10-20 parallel PCR reactions. Each PCR reaction contained 25 μL of NEB HF 2× mix, 1 μM of forward (5′-GCCATCCAGAAGACTTACCG-3′) and 1 μM of reverse (5′-GCAGTCGT-GAAGACGGAAAC-3′) primers, 1.5 ng of DNA oligo template, and water to a final volume of 50 μL. The thermocycler conditions were 98 °C for 1 min; 6-10 cycles of 98 °C for 15 s, 62 °C for 15 s, 72 °C for 15 s; final extension at 72 °C for 1 min. The generated amplicons were purified using the Monarch PCR DNA Cleanup Kit (New England BioLabs) according to manufacturer's DNA cleanup and concentration protocol and subsequently used for Golden Gate Assembly. Ten parallel reactions were prepared per pool, each with 1 ng purified sgRNA amplicon, 212 ng of vector, 5U BbsI (NEB), 200U T4 DNA ligase (NEB), 10× ligase buffer (NEB) and water up to 20 μL. Reactions were incubated in a thermocycler with the following protocol: 100 cycles of 37 °C for 5 min and 16 °C for 5 min; 65 °C for 15 min; hold at 4 °C. Ligations were pooled and purified using the Monarch PCR DNA Cleanup Kit (New England BioLabs) and electroporated into Endura ElectroCompetent

Cells (Lucigen) using a Bio-Rad MicroPulser. After a recovery period of 1 h at 32 °C, bacteria were inoculated in LB medium with ampicillin (100 μg/ml) and expanded for <12 h at 32 °C. The following day, cultures were spun down and plasmid DNA was extracted using the Qiagen Plasmid Maxi Kit (Qiagen). In order to calculate library coverage, a dilution series of bacteria was also prepared after the recovery period—bacteria were plated in LB-agar-ampicillin and colonies were counted the next day. We estimated at least 2000 colonies per sgRNA for both pools of the YMCi-160K library and at least 6000 colonies per sgRNA for the YMCi-13K library. We performed a quality control of both libraries by next-generation sequencing (HiSeq2500, Illumina), which retrieved >99% of the sgRNAs present in the YMCi-160K and YMCi-13K libraries and showed a homogeneous distribution of all sgRNAs. Raw counts for the two libraries are available in Supplementary Data 2 and Supplementary Data 6.

## Pooled CRISPR screening
The lentiviral pools of the YMCi-160K and the small-scale validation library YMCi-13K were transduced into NCI-H2052, MSTO-211H, 92.1 and MEL202 CRISPRi dCas9/KRAB-expressing cells at a multiplicity of infection of <0.3 and maintained as two technical duplicates at >500× library representation each. Transduction efficiency was determined 2 days post-infection by flow-cytometric analysis of RFP+ cells and puromycin (Invitrogen) was added at 1.5 μg/mL. Cells were routinely split to maintain sub-confluency levels at all times throughout the screen while ensuring maintenance of library representation until day 22, when samples were taken for gDNA extraction. Genomic DNA was extracted using the Blood and Cell Culture DNA Maxi Kit (Qiagen). Samples were randomized using alphanumerical IDs for subsequent processing. Library for next-generation sequencing were prepared and analyzed as previously reported[28].

## Gene expression analyses (qPCR)
For mRNA expression analysis, total RNA was extracted from cells using RNeasy Plus Mini kit (Qiagen). Real time PCR was performed on Quant Studio 6 (Thermo Fisher) using iTaq Universal Probes One-Step Kit (Biorad) according to manufacturer's instruction. Expression was normalized to *human* GAPDH. The PrimeTime Predesigned qPCR Assays from Integrated DNA Technologies (IDT) can be found in Supplementary Data 9.

## ChIP-seq
ChIP-seq was performed as previously described[6]. Briefly, cells were cross-linked in 1% formaldehyde for 10 min at room temperature after which the reaction was stopped by addition of glycine to a final concentration of 0.125 M. Cells were harvested in SDS Lysis Buffer (100 mM NaCL, 50 mM Tris-HCl pH8.0, 5 mM EDTA pH8.0, 0.02% NaN3, 0.5% SDS), pelleted and resuspended in ice-cold ChIP-buffer (100 mM NaCL, 75 mM Tris-HCl pH8.0, 5 mM EDTA pH8.0, 0.02% NaN3, 0.25% SDS, 2.5% Triton X-100). Chromatin was disrupted by sonication using an EpiShear sonicator (Active Motif) to obtain fragments of average 200 to 500 bp in size. Suitable amounts of chromatin extracts were incubated for 16 h with primary antibodies (YAP1 Abcam cat. ab52771 3:1000, TAZ (V386) CST cat. 4883 5:1000, TEAD4 Abcam cat. ab58310 3:1000, PanTEAD CST cat. 13295 8:1000, Rpb1 NTD CST cat. 14958 5:1000, H3K4me1 CST cat. 5326

5:1000, H3K4me3 Millipore cat. 07-473 1:1000, H3K27ac CST cat. 8173 5:1000, Supplementary Data 9). Immunoprecipitated complexes were recovered using Protein G Dynabeads (Invitrogen), and DNA was recovered by reverse cross-linking and purified using SPRIselect beads (Beckman Coulter). Libraries for ChIP-seq were generated using the Ovation Ultralow Library System V2 (NuGEN), and barcodes were added using New England Biolabs (NEB) Next Multiplex Oligos for Illumina (NEB, Index Primers Set 1) according to the manufacturer's recommendation. The antibodies used for ChIP-seq are provided in Supplementary Data 9.

## Cut&Tag

Cut&Tag (Cleavage Under Targets and Tagmentation) assay was performed as previously described[56]. Briefly, 400,000 cells (or 40,000 cells per histone mark) were harvested, washed and mixed with activated concanavalin A-coated magnetic beads (Polysciences) at room temperature, for 10 min. The bead-bound cells were then suspended in 100 μl of Dig-wash buffer containing 2 mM EDTA, 1% BSA and primary antibody (TEAD4 Abcam cat. ab58310 1:500, JUN BD Biosciences cat. 610326 1:500, SOX10 CST cat. 89356 1:500, MITF CST cat. 97800 1:500, TFAP2A Sigma cat. HPA028850 1:500, CTCF CST cat. 3418 1:500, H3K27ac CST cat. 8173 1:500, Rabbit IgG Antibodies-online cat. ABIN101961 1:500, Mouse IgG H&L Abcam cat. ab46540 1:500, YAP1 Abcam cat. ab52771 1.5:500, FOSL1 Abcam / VWR cat. ab252421 1.5:500, PAX3 Millipore/sigma cat. HPA063659 2.5:500, see Supplementary Data 9) and incubated overnight at 4 °C. The secondary antibody was added to cells and incubated at room temperature for 1 h. After washing, 50 μL of pA-Tn5 adapter complex was added and incubated at room temperature for 1 h. Cells were washed, resuspended in tagmentation buffer (300 μl) and incubated at 37 °C for 1 h. To stop tagmentation and solubilize DNA fragments, stop buffer was added delivering EDTA (15.8 mM), SDS (0.1%) and Proteinase K (0.16 mg/mL), and samples were incubated at 55 °C for 1 hour. DNA was then extracted using the Phenol-Chloroform extraction method. For PCR amplification with indexing primers, the following thermocycler program was used: 72 °C for 5 min, 98 °C for 30 s, 14 cycles of 98 °C for 10 s and 63 °C for 10 s, final extension at 72 °C for 1 min, and hold at 4 °C. Libraries were purified with 1.3 volumes of SPRIselect beads (Beckman Coulter, B23318) and library size distribution was checked using High Sensitivity D1000 ScreenTape and reagents (Agilent). Paired-end Illumina sequencing was carried out on a NovaSeq 6000 instrument (Illumina). The antibodies used for Cut&Tag are provided in Supplementary Data 9.

## HiC and H3K27ac-HiChIP

The HiC and HiChIP experiments were performed in NCI-H2052 and 92.1 cells using the Arima-HiC Kit (Arima Genomics, A510008) or the Arima-HiC⁺ Kit (Arima Genomics, A101020) for mammalian cell lines, following manufacturer's instructions. Briefly, chromatin was cross-linked and digested using a restriction enzyme cocktail. The 5′-overhangs were filled-in with a biotinylated nucleotide and then ligated. At this stage, the HiC samples were purified, fragmented, enriched by biotin pull down and the enriched fragments were used to prepare a custom library following instructions on the Arima user guide for HiC Library Preparation using KAPA Hyper Prep Kit (Roche). After ligation, the HiChIP chromatin samples were sheared, bound to an antibody recognizing H3K27ac (Cell Signaling Technology, 8173), immunoprecipitated and purified. The resulting fragmented DNA molecules that were marked by H3K27ac were then enriched for biotin-labeled fragments and library preparation was done following instructions on the Arima user guide for HiChIP Library Preparation using Swift Biosciences Accel-NGS 2S Plus DNA Library Kit (Swift Biosciences/IDT). The HiC and HiChIP libraries were sequenced in a NovaSeq 6000 (Illumina) using paired-end mode. Raw data have been deposited to SRA with BioProject ID: PRJNA949402.

## Computational analyses

All NGS data were initially processed in a blinded fashion by data analyst using randomized alphanumerical ID assigned to samples. When analyses required specific contrasts, the nature of the samples were revealed to investigators. No statistical method was used to predetermine sample size and no data were excluded from the analyses.

**ChIP-Seq data analysis.** The analysis of ChIP-Seq data was performed using a modified version of the snakePipes pipeline[57]. Briefly, the FASTQ files were aligned to the hg38 *human* reference genome using Bowtie2. Duplicate reads were marked using picard MarkDuplicates and were removed using samtools (samtools view -F 1024). RPKM normalized coverage (bigwig) files were generated using deepTools bamCoverage. Peakcalling was done using MACS2 (default parameters). The R package DiffBind was used to make consensus peaksets and do differential peakcalling. For the heatmaps showing YAP occupancy at consensus peak sites (Fig. 1e) and ChIP-seq correlation heatmaps (Figs. 1c and S1C), the consensus peakset between the plotted samples was derived using the DiffBind package. The dba.count function was used to obtain input subtracted ChIP-Seq read counts in the consensus peaks for each sample (using the score = DBA_SCORE_TMM_MINUS_FULL argument). For Figs. 1c and S1C, the correlation of read counts at the consensus peaksets between the samples was calculated using the dba.plotHeatmap function. The resulting correlation matrix was used for hierarchical clustering of the samples. The correlation heatmap and associated dendrogram were plotted using the ggplot2 and ggdendro packages. For Fig. 1e, the YAP1 peaksets from the four cell lines were grouped by disease of origin and the YAP1 peaks were sorted by the log2-fold change in YAP1 occupancy between the uveal melanoma and mesothelioma cell lines. The heatmap was produced using ggplot2. ChIP-Seq occupancy heatmaps (Figs. 1d and S1D) were plotted in a 4 kb window around the YAP1 peak summits (+/− 2 kb from the summits). The 4 kb window was divided into 100 bins and the occupancy in each bin for every YAP1 peak was calculated using the ScoreMatrixList function from the R package genomation. The signal within each factor was normalized such that: (1) The occupancy for the top two percentile of bins were set to the 98th percentile, (2) The occupancy for the bottom one percentile was to the first percentile, and (3) The signal was scaled between zero and one. The YAP1 peaks were clustered based on Rpb1, H3K4me1 and H3K4me3 signal using k-means clustering ($k = 2$). The peaks within each cluster were sorted by the total YAP1 signal within each peak. The heatmaps were plotted using the ggplot2 package. Raw data have been deposited to SRA with BioProject ID: PRJNA949402.

**Cut&Tag data analysis.** The Cut&Tag data was analyzed using the same pipeline and same parameters as the ChIP-Seq data except for the peakcalling stage. The peakcalling was done using a modified MACS2 command: macs2 callpeak -t input_file -p 1e-5 -f BEDPE -n output_name[56].

ChIP-seq/Cut&Tag signal coverage was calculated from bigwig files in a window of ±250 bp using Genomation (v. 1.26.0[58]) using regions identified as hits from CRISPR screens (for Figs. 5b and 6b) or the union of YAP peaks (for heatmap in Fig. S9F) as reference. For Fig. S9F, Log-transformed signal was used to calculate pairwise Spearman correlation among samples. Raw data have been deposited to SRA with BioProject ID: PRJNA949402.

**Pooled CRISPR screens data analysis.** Sequencing reads were aligned to the sgRNA library. For each sample, sgRNA reads were counted. Results from individual samples were scaled for library size and normalized using the trimmed mean of M-values (TMM) method available in the edgeR Bioconductor package[59]. The Log₂ fold change, $p$ values and FDR values for individual sgRNAs were obtained by fitting

a quasi-likelihood negative binomial generalized log-linear model to the count data using the edgeR package. The log fold change values of different sgRNAs targeting the same YAP1 peak summit (summit-level analysis) were aggregated using the MaGeCK tool. Annotation of the gene transcription start sites (TSS) closest to an sgRNA or YAP1 peak summit was done using HOMER. EdgeR and MaGeCK analyses are provided in Supplementary Data 3 and Supplementary Data 4, respectively. For Fig. S3A, only control sgRNAs that are common to both pools were selected for plotting. The log2-fold change values for each of the control sgRNAs in the two pools were plotted using the ggplot2 package. For Fig. S3B, only experimental guides targeting YAP peaks are plotted. TMM normalized counts for each replicate were log2 transformed after addition of a pseudocount (equal to the lowest non-zero TMM normalized count). The scatter plot was made using ggplot2 and the points colored according to their density to enable identification of areas with a high number of overlapping points. The density was calculated by kernel density estimation using a two-dimensional gaussian kernel.

For analysis of YMCi-13K, we calculate the relative fitness scores of sgRNAs as previously described[28]. Briefly, we first normalize median counts of control sgRNAs to 1000 and other sgRNAs proportionally to correct for sample variations. We then aggregate sgRNAs that target the same locus and fit the time-series normalized counts to an exponential model to calculate the relative fitness score (over the control group) using the following equation.

$$x_t^i = x_0^i \times e^{\alpha^i \times t} \quad t = (8, 15, 22) \tag{1}$$

where $x_t^i$ denotes the normalized count of the $i^{th}$ target at time $t$ and $x_0^i$ represents the normalized count in original libraries. $\alpha^i$ denotes the relative fitness score of this target. A negative value represents cell growth inhibition and a positive one represents promotion of cell proliferation. We solve the equation by transforming it with natural log link function and get,

$$\ln\left(x_t^i\right) = \ln\left(x_0^i\right) + \alpha^i \times t \quad t = (8, 15, 22) \tag{2}$$

We then estimate the relative fitness score with linear regression model from Statsmodels[60] and visualize both relative fitness scores and $p$ values for each target group in Fig. 3a. Results from the fitness score analysis of YMCi-13K are provided as Supplementary Data 7.

**HiChIP and HiC data analysis.** The HiChIP analysis was done using the MAPS pipeline for ARIMA HiChIP pipeline https://github.com/ArimaGenomics/mapping_pipeline using the default parameters. The peakcalling files output by the MAPS pipeline were converted into the bigInteract format and used for visualization of contact loops on the genome browser. HiC data were processed using the DCC-HiC ENCODE pipeline with default parameters[61]. Data were converted into mcools files for visualization using HiGlass[62].

**Transcription-factor motifs analysis.** Transcription-factor motifs analysis in Fig. 2d was performed using MonaLisa (v 1.2.0)[63]. Briefly −Log10(ΔRRA score) from Mageck (Supplementary Data 4) was calculated comparing the day 22 results for NCI-H2052 vs. 92.1. YAP summits were then binned in 6 bins of equal size and MonaLisa was used to query JASPAR2020[64] for TF motifs enriched in ±250 bp distance from the peak summit. TF motifs showing -Log10(Adj. *P* value) > 0.6 in the extreme bins were retained for plotting.

**Public large-scale genomics datasets.** Chronos score version 22Q1 were obtained from DEPMAP (www.DEPMAP.org). Models were divided according to lineages Mesothelioma, Uveal Melanoma and Others. Log2CPM values for TCGA expression were downloaded from GDC (GDC (cancer.gov) and, similarly, cases were subdivided according to

three categories. Statistical significance between MPM and UM models/cases was assessed with Kolmogorov–Smirnov test.

### Reporting summary
Further information on research design is available in the Nature Portfolio Reporting Summary linked to this article.

## Data availability
The ChIP-seq, Cut&Tag, HiC and H3K27ac HiChIP sequencing data generated in this study have been deposited to SRA with BioProject ID: PRJNA949402 and can be publicly accessed here ID 949402 - BioProject - NCBI (nih.gov). All other data are available in main text, Supplementary information files or source data files. Source data are provided with this paper.

## Code availability
Computational analyses have been performed using open-source code as indicated in the "Methods" section. No proprietary code/software have been employed.

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

## Acknowledgements

We thank Bruno Amati for critical reading of the manuscript and members of the Schübeler lab for insightful discussion. We are additionally indebted to several members of Oncology Disease Area of NIBR for sharing information and providing valuable input. D.S. acknowledges support from the Novartis Research Foundation, the Swiss National Science Foundation (no. 310030B_176394) and the European Research Council under the European Union's Horizon 2020 research and innovation program grant agreements (nos. ReadMe-667951 and DNAaccess-884664).

## Author contributions

I.A.M.B., T.M., R.L. and G.G.G. performed molecular and cellular biology experiments; R.G., S.M., T.P.M., S.W., C.S., F.J., K.S., L.B. and G.G.G. performed bioinformatic analyses; U.N., J.K., M.A., A.L., C.R. generated NGS data; I.A.M.B., J.G., L.T., D.S., T.S. and G.G.G. conceptualized and supervised the project. The manuscript was written by I.A.M.B. and G.G.G. with input from L.T., D.S. and T.S.

## Competing interests

All the authors affiliated with Novartis Institutes for Biomedical Research are shareholders and/or employees of Novartis. The remaining authors declare no competing interests.
