## [Peer Review File · Nature Communications]

REVIEWERS' COMMENTS

Reviewer #1 Remarks to the Author

This is a review of NCOMMS-22-33613-T "Cancer lineage-specific regulation of YAP responsive elements revealed through large-scale functional epigenomic screens" by Barbosa, Galli, and colleagues.

This work presents a comprehensive analysis on YAP/TAZ enhancer response elements, and supports several prior observations (that did not have firm proof) that YAP/TAZ regulating enhancers are often distal from the transcription start site and has a distinct genomic signature linked to cellular lineage. This work has many strengths, including the potential impact on considering cell type for YAP/TAZ/HIPPO based therapeutics, as well as a broader scope on how this pathway functions in lineage specificity in tumorigenic and possibly normal cell types. The research strategy is highly comprehensive, and well-controlled in experimental groups, bioinformatics, statistics, and multiple supportive methodologies. In several places in the manuscript, findings are validated through other approaches. The supplemental datasets are detailed and supportive of this work, and will provide a foundation for the field. I believe this work is highly impactful, with only moderate to minor deficiencies.

Review points

Moderate points

Within the paper there is some generalization in terms of the uveal melanoma lines, for example the heading "YREs rewire a network of melanoma-specific master regulators." Melanoma subtypes have very different molecular signatures, with uveal melanoma often driven by G protein signaling versus other types of melanoma, ie cutaneous melanoma with a heavy reliance of the BRAF/RAS pathway. It is recommended that the more specific designation of "uveal melanoma" be used throughout the manuscript.

It is suggested that some of the supplemental data be moved to the main manuscript body. This includes Figure S4E, since the identification of specific and distinct binding motifs is scientifically impactful and supports the main focus of the work, ie lineage specificity.

The investigators perform a validation experiment, demonstrating lineage specificity genes between mesothelioma and uveal melanoma cells, with a focus on specific genes (NRSF2, NR4A3, KEAP1, etc). This is a strong validation experimental strategy. However, this experiment as presented in Figure S6C,D is missing some essential controls. For example, there should be some confirmation that the targeted gene is indeed repressed (ie on a transcript or protein level), and there should be inclusion of negative controls as well.

In Figure 3, it is not clear how the distance from the TSS in Figure 3B was determined, due to complicating issues of overlapping and neighboring genes. The supplemental data suggest that this may be calculations of a peak's distance to the nearest defined TSS. A refreshing addition to this paper was the use of HiC, rather than an implied connection of an enhancer to a gene due to proximity. Indeed, the HiC data presented here supports that YAP/TAZ enhancers are not always affiliated with the closest gene. Some clarity for how the analysis shown in Figure 3B is suggested. In addition, how were the loci defined in Figure 3C? Are these the closest gene loci or linked to the HiC data?

The data in Figures 4-6 match up HiC data to CUT&AG or ChIPseq data, and is a strength of the paper. It is not clear, however, how the peaks identified were matched to the HiC data. While the interpretation of these data by the investigators is logical and a likely possibility, it may be argued that these are distal enhancers for neighboring genes. There are some fitness scores in the figures, some clarification of how these were calculated and datasets compared would be beneficial.

There are some different FDR cutoff ranges mentioned (ie, line 124, Figure S2, methods page 10, lines 367). It is not clear why these cutoff ranges were selected and why there is different FDR cutoffs between different cell lines. Some clarification in the methods is suggested.

There is some concern for overgeneralization, such as in conclusions on lineage specificity based on analysis of only two, and often only one, cell type of each cancer group. Further, the schematic shown in Figure 6 involving lineage specific transcriptional groups is very interesting, but may be a stretch based on the data presented here. However, concerns here are reduced due to the overall comprehensive experimental approach employed throughout the manuscript.

Minor points

There are some minor typographical errors in the document, including "T systematically interrogate (the) Yap cistrome" (line 36), "released into (the) nucleus" (line 54), and "derived network of melanocytic transcription factors" (line 82). Overall, the manuscript is well written and with enough scientific description in most of the document.

There are some formatting errors with references, such as Kaya-Okur et al 2019 (lines 443, 500).

The link listed in line 536 is no longer an active website and leads to a 404 page.

Comment

This work brings up several very interesting scientific questions. For example, are YAP/TAZ involved directly in lineage specificity, or are recruited by lineage specifying factors and promote or modulate the expression of these genes? This is an interesting potential following the schematic presented in Figure 6. What these factors are specifically doing in terms of gene expression is not understood, and this work provides some clues.

Reviewer #2 Remarks to the Author

Barbosa et al. present a detailed and functional analysis of YAP target genes in two cancer types that are driven by YAP activation, mesothelioma and uveal melanoma. They found that the target gene programs are different between the two cancer types. Surprisingly, they found that YAP-TEAD interaction is required only in mesothelioma but not uveal melanoma, indicating that YAP acts through different mechanisms in the two cancer types. The study is well executed and the conclusions largely follow from the data. The results are important as they inform therapeutic strategies and potential use of TEAD inhibitors to treat cancer. However, the authors should add analyses of YAP target gene engagement under TEAD inhibitor treatment and combined TEADi and YAP knockdown to fully support their conclusions.

1. Because TEADs can act as suppressors in the absence of YAP, the authors need to perform a combination treatment of TEADi and YAP knockdown in order to show that the effects of YAP knockdown in uveal melanoma do not require TEAD. Without this experiment the results are not conclusive.

2. Do YAP target sites have TEAD motifs and are they bound by TEADs (cut&tag)? This is especially important to know for the lineage-specific sites.

3. What is the effect of TEADi on YAP occupancy at YREs? I am surprised that the authors did not include such analysis as it would answer one of the most straight forward questions resulting from their model that YAP can act independent of TEAD in uveal melanoma.

Minor comments

1. line 133: the two library pools are not well described

2. How far on the DNA does KRABi exhibit inhibitory effects and how does this affect the conclusions?

3. line 282: "silencing" is not the correct term here. Better "activity" or rephrase

Reviewer #3 Remarks to the Author

In their manuscript entitled "Cancer lineage-specific regulation of YAP responsive elements revealed through large-scale functional epigenomic screens", Barbosa et al. study functional differences downstream of YAP in two distinct cancer types. They choose malignant pleural mesothelioma (MPM) and uveal melanoma (UM) as prototypes for driver mutations in/outside the canonical Hippo pathway as well as TEAD dependency/independency respectively.

To identify relevant target genes downstream of YAP in both cancer types, the authors perform a CRISPRi screen recruiting a dCas9-KRAB fusion to YAP binding sites identified in a ChIP experiment. These screens and validation thereof is well-executed and delivers high confidence target loci with differential fitness effects upon KRAB recruitment.

The authors perform an impressive amount of epigenetic analyses to map out binding sites of YAP, TAZ, TEAD, as well as various histone marks and also analyze chromatin structure by HiC. The manuscript further contains e.g. various experiments using chemical inhibitors, overall a true tour de force. They can clearly correlate sites of cell line specific YAP binding and associated histone marks to sites reducing cellular fitness upon KRAB recruitment. Further, they convincingly demonstrate proximity of these regions to the transcriptional start sites of respective genes by HiC. As example, they identify cell type specific enhancer elements at the MYC locus for the studied cell lines. (Of note, in Figure S7B only one of the two +1800 sgRNA in the validation scores in 92.1 at a time scale that would be expected for the MYC locus and effects on MYC expression are overall small. Repeating this experiment with further sgRNAs scoring in the initial screen would thus increase trust in the result.)

The authors go on to demonstrate lineage specific cooperativity with other signaling pathways for MPM and UM respectively. While the MAPK responsive TFs JUN, FOSL1 and FOSB appear to be specifically required in MPM and silenced upon recruitment of KRAB, MITF, SOX10 and PAX3 are only essential in UM. As for differentially active enhancers in the MYC locus such result does not come at great surprise given known genetic dependencies and mechanisms of e.g. resistance to targeted therapies.

In these sections about differential interactions with other TFs however data display becomes incomplete and inconsistent. The authors refer to Fig. 3C for lineage specificity, however the resolution here does not allow for full evaluation. Fig 5A, 6A, S8A, and S10C lacks tracks for the KD experiment of the other cell lines respectively. Moreover, validations by competition assay are done only selectively (e.g. for FOSL1 in S9A), and expression analysis is also not consistent with competition assays but rather only displayed in responder cell lines (S9B, S10B). Please provide a more coherent validation and support of your claims based on the methods well-established in your labs.

The authors start the analysis based on their mapping of YAP binding sites to then correlate epigenetic features to the ability to silence with CRISPRi. Their studies culminate in a model (Fig. 6C) whereby YAP together with lineage specific transcription factors induces target genes and thus recruitment of KRAB to that locus results in reduced cellular fitness. However, the authors only show that KRAB reduces fitness if bound to these sites, so in other words they show these sites to be accessible and likely functional enhancers. They provide no functional evidence that YAP indeed induces expression in said manner and at that locus, it may just bind the open enhancer in a cell type specific manner. While I agree that the correlative evidence is strong, functional claims are not really supported by data. An alternative and plausible interpretation of the data could e.g. be that Cas9 binding/KRAB is particularly effective if recruited to accessible loci (ChIP) in proximity (HiC data) to the TSS. In this context I may mention that I do not find figures 5B and 6B particularly convincing, as the data are strongly overlapping between the three groups and minimal differences could be explained by enhancer accessibility.

If the authors wish to maintain the claim visualized in the model, functional data are required at

least for some examples. Various experiments could support the claim functionally such as ChIP or HiC upon acute loss of YAP, acute overexpression, or recruitment of YAP to lineage specific enhancers e.g. via dCas9. Locus specific degradation of YAP could also be envisioned. Alternatively, YAP binding motifs within identified regions could be targeted with WT Cas9 or better by induction of point mutations or HR to reach a higher resolution in the genetic perturbation and confirm results independent of KRAB. (In this case care must be taken that a transient DNA damage response does not result in false positive results.) I would consider such functional experiments critical in light of the claims of the manuscript, but would leave the decision of how to probe functionality to the authors.

The figures are of remarkable good quality and offer a very consistent layout, which I would like to point out as an outstanding feature of this manuscript, congratulations! Similarly, the manuscript is well structured, very well written, and overall very mature.

Reviewer #4 Remarks to the Author

In this study, the authors employed large-scale functional epigenomic screens of YAP regulatory elements (YRE) in both MPM and UM, and revealed unanticipated lineage-specific features of the YAP regulatory network that provide important insights to guide the design of tailored therapeutic strategies to inhibit YAP signaling across different cancer types. The manuscript is well-organized, well-written, and more valuable, generating a large amount of functional screening data. The reviewer has several suggestions:

1. The authors focused on cancer-specific patterns, especially in MPM and UM, for example, figure 1B – is it possible to examine the pattern in other cell lines of different cancer types? This may be completed through public datasets, such as GDSC and CTRP, which cover thousands of cell lines with hundreds of drugs – if the TEADi is included in the drug screening.
2. Similar to the above point, the authors may also consider expanding the scope by utilizing public data, such as DepMap for CRISPR screening result in different cancer cell lines, if applicable.
3. The authors generate a large amount of data, which will be valuable resources for the research community. The authors should release the related data, not only the HiC and Hi-ChIP (deposited in GEO but without the GSE number) but other data, including pooled CRISPR screening data, ChIP-seq, and Cut& Tag data.

Reviewer #5 Remarks to the Author

In this manuscript, Barbosa et al. present a study of the YAP transcriptional co-activator in the context of two cancers: malignant pleural mesothelioma (MPM) is driven by mutations in components of the Hippo signaling pathway, which is responsible for suppressing the nuclear activation of YAP, while in uveal melanoma (UM), YAP is activated by oncogenic mutations in GNAQ/GNA11 GTPases, independent of the Hippo pathway. The authors utilize genome-wide functional epigenomic screens to reveal YAP-responsive elements (YREs) shared by and specific to each cancer lineage.

A comprehensive analysis is performed here utilizing a wide-range of assays including CRISPR-based YMCi libraries, ChIP-Seq of an array of epigenetic marks and HiC/HiChIP to look at chromosome conformation. Using this platform, the authors systematically interrogate YAP-responsive cis-regulatory elements in a genome-wide fashion and present novel insights about lineage-specific mechanisms involved in YAP-dependent cancers. Their findings suggest shared and lineage-specific programs operating in the two cancers, which provide guidance for development of selective therapeutic strategies in YAP-related cancers.

I believe this is an excellent study and the authors have done a great job summarizing their findings succinctly and providing suitable context for the reader. I have some minor comments

predominantly regarding data visualization.

Specific comments:

1. Page 4, line 124: It is not clear from the main text, how the top 18063 selected YAP-peaks are distributed in terms of lineage-specificity. This information should be highlighted here instead of just in supplementary figures.
2. Figure 2B: This figure is hard to understand. The spread of points in pan-lethal and experimental groups look very similar, but the violin plots are very different. Why is this? Also, how are 'experimental' groups defined? Are these all 18000 selected regions? I could not find this information in figure legends or in methods.
3. Page 4, line 143-146: "Within experimental sgRNAs, as previously reported for Estrogen Receptor, only a small fraction of perturbations displayed significant scoring both in UM and MPM, suggesting that only a subset of YREs is necessary for cell proliferation (Figure 2B)."
It is very difficult for the reader to make this conclusion from Figure 2B. Either the figure legend needs to be more descriptive, or this sentence needs to include more information.
4. Page 4, line 149: Even though the authors analyzed three different time-points: 8, 15 & 22 days post-infection, only day 22 is analyzed here. The authors should explicitly note the reason behind this choice here.
5. Figure 4B: The 'UM-specific' +1.8Mb region appears to have strong HiC signal in the MPM cell line, suggesting physical interactions with the MYC locus. There also appears to be some H3K4me1 CTCF signal at this locus in this cell line. Even though there is no H3K27Ac signal, the above appear to indicate that the structure of this particular region is not lineage-specific. The reverse is not true of the UM-cell line. How do the authors explain this?
6. Page 6, line 209-211: This claim is not very well supported by the figure. The differences appear to be varied and it's hard to collate the p-values from all the different groups shown in the figure. The authors should consider depicting this information using a table.
7. Figure 5C: What is the meaning of the 'score: ...' numbers at the top right of the figure. This is not described in the figure legend.
8. Correlation heatmaps depicting positive values ranging between 0 and 1 are shown using divergent colormaps, e.g. Fig 1C, S1B, S5C, S9C. This type of colormap is visually misleading as values in the middle of the scale are shown in white and effectively highlighted, while values at the bottom of the scale (with 0 correlation) also appear unduly strong. These figures should really use a sequential colormap, e.g. viridis.

Minor comments:

1. Page 8, Line 274: AP-1 motifs "flanking" TEAD motifs

RESPONSE TO REVIEWERS' COMMENTS

Reviewer #1 (Remarks to the Author): expertise in transcription factor signalling in melanoma

This is a review of NCOMMS-22-33613-T "Cancer lineage-specific regulation of YAP responsive elements revealed through large-scale functional epigenomic screens" by Barbosa, Galli, and colleagues.

This work presents a comprehensive analysis on YAP/TAZ enhancer response elements, and supports several prior observations (that did not have firm proof) that YAP/TAZ regulating enhancers are often distal from the transcription start site and has a distinct genomic signature linked to cellular lineage. This work has many strengths, including the potential impact on considering cell type for YAP/TAZ/HIPPO based therapeutics, as well as a broader scope on how this pathway functions in lineage specificity in tumorigenic and possibly normal cell types. The research strategy is highly comprehensive, and well-controlled in experimental groups, bioinformatics, statistics, and multiple supportive methodologies. In several places in the manuscript, findings are validated through other approaches. The supplemental datasets are detailed and supportive of this work, and will provide a foundation for the field. I believe this work is highly impactful, with only moderate to minor deficiencies.

We thank the reviewer for appreciating our work and providing valuable comments to strengthen our manuscript both from the conceptual as well as technical standpoint.

Review points

Moderate points

Within the paper there is some generalization in terms of the uveal melanoma lines, for example the heading "YREs rewire a network of melanoma-specific master regulators." Melanoma subtypes have very different molecular signatures, with uveal melanoma often driven by G protein signaling versus other types of melanoma, ie cutaneous melanoma with a heavy reliance of the BRAF/RAS pathway. It is recommended that the more specific designation of "uveal melanoma" be used throughout the manuscript.

We thank the reviewer for spotting this inaccuracy, we corrected the specific designation of "uveal melanoma" throughout the manuscript.

It is suggested that some of the supplemental data be moved to the main manuscript body. This includes Figure S4E, since the identification of specific and distinct binding motifs is scientifically impactful and supports the main focus of the work, ie lineage specificity.

We thank the reviewer for the suggestion. We moved figure S4E to main figure 2D. We then added additional volcano plots for differential scoring of the primary YMCi screen in figure S4D according to the recommendation from reviewer #5 (see Figure 19 for Reviewers in reviewer #5's section).

The investigators perform a validation experiment, demonstrating lineage specificity genes between mesothelioma and uveal melanoma cells, with a focus on specific genes (NRSF2, NR4A3, KEAP1, etc). This is a strong validation experimental strategy. However, this experiment as presented in Figure S6C,D is missing some essential controls. For example, there should be some confirmation that the targeted gene is indeed repressed (ie on a transcript or protein level), and there should be inclusion of negative controls as well.

We thank the reviewer for appreciating our experimental strategy and we do acknowledge that we should extend our validation. Thereby in supplementary Figure S6, we now include extensive validation by transducing all the four cell lines employed throughout the study with two control sgRNAs, three sgRNAs targeting YAP and two sgRNAs targeting lineage specific genes or enhancers. We evaluated both phenotypic effects (by time-resolved flow-cytometry competition assay) and target gene regulation by qPCR.

These new data validate that all four cell lines are sensitive to YAP silencing and canonical target genes are dependent on YAP in each model. See FIGURE 1 for reviewer below. These data have now added to Figure S6 A and B of the manuscript.

FIGURE 1 for reviewers

We could additionally validate in the cell lines not analyzed in the first version of the manuscript, that lineage-specific hits are indeed selective dependencies and that the employed sgRNAs are effective in silencing the related gene. See FIGURE 2 for reviewers below. These data have been now included in supplementary figure S6 C-E.

FIGURE 2 for reviewers

In Figure 3, it is not clear how the distance from the TSS in Figure 3B was determined, due to complicating issues of overlapping and neighboring genes. The supplemental data suggest that this may be calculations of a peak's distance to the nearest defined TSS. A refreshing addition to this paper was the use of HiC, rather than an implied connection of an enhancer to a gene due to proximity. Indeed, the HiC data presented here supports that YAP/TAZ enhancers are not always affiliated with the closest gene. Some clarity for how the analysis shown in Figure 3B is suggested. In addition, how were the loci defined in Figure 3C? Are these the closest gene loci or linked to the HiC data?

Figure 3B represents the distance of scoring sgRNAs to the closest TSS (as depicted on the x-axis). The reason for such a choice, even with Hi-C and Hi-ChIP data in hand, is due to the limitation of resolution intrinsic to these technologies. Indeed Hi-C and Hi-ChIP are unable to exhaustively detect loops under

approximately 50Kb in linear distance. Thereby, we preferred to use an unbiased/consistent metric such as distance to closest TSS.

For Figure 3C, we inspected the loci of every hit in our screen for evidence of loops to a protein coding gene and annotated the figure with such information.

We clarified these points in the figure legend and provide details in Supplementary Table 8, which we missed to reference in the first version of our manuscript. This table includes both the annotation to closest TSS as well as the putative TSS, annotated upon inspection of Hi-C/Hi-ChIP loops for all hits shown in heatmap of Figure 3C.

In FIGURE 3 for reviewers below is the example for hits found within the MYC locus, taken from Supplementary Table 8:

- Columns in green: order of appearance in heatmap and scoring category to which the hit belongs;
- Columns in grey: genomic coordinates of YAP-peak summits and YMCi-library Union IDs;
- Columns in blue: annotation of closest TSS and distance to closest TSS (if the same TSS is annotated more than once it will be given a number – 1, 2, 3, ..., n);
- Columns in orange: putative target assigned to screen-hits, based on Hi-ChIP and Hi-C analysis, and distance to TSS of putative target gene.

Heatmap_order _Figure3	Category _score	chr	absolute summit	UnionID	UnionID_AbsSummit	Closest_TSS	Closest_TSS_nr	Distance_to closest_TSS	Putative_target _gene	Distance to TSS of putative target
49	Common	chr8	127736216	000125805	000125805_127736216	MYC	MYC.2	89	MYC	TSS
94	Common	chr8	127737154	000125805	000125805_127737154	MYC	MYC.3	849	MYC	TSS +0.85Kb
104	Common	chr8	127737903	000125805	000125805_127737903	MYC	MYC.4	1734	MYC	TSS +1.7Kb
109	Common	chr8	127735343	000125805	000125805_127735343	MYC	MYC	-10	MYC	TSS
123	Common	chr8	127926373	000125823	000125823_127926373	TMEM75	TMEM75.6	22147	MYC	TSS +190Kb
142	Meso	chr8	128176808	000125861	000125861_128176808	MIR1208	MIR1208.3	26655	MYC	TSS +440Kb
153	Meso	chr8	128183353	000125861	000125861_128183353	MIR1208	MIR1208.5	33457	MYC	TSS +447Kb
158	Meso	chr8	128167778	000125861	000125861_128167778	MIR1208	MIR1208	17822	MYC	TSS +430Kb
180	Meso	chr8	127908035	000125821	000125821_127908035	TMEM75	TMEM75.2	40638	MYC	TSS +171Kb
252	Uveal	chr8	129552182	000126013	000126013_129552182	CCDC26	CCDC26	22780	MYC	TSS +1800Kb
253	Uveal	chr8	127735674	000125805	000125805_127735674	MYC	MYC.1	-45	MYC	TSS
263	Uveal	chr8	127734159	000125805	000125805_127734159	CASC11	CASC11	-222	MYC	TSS -2Kb
266	Uveal	chr8	128995012	000125947	000125947_128995012	LINC00976	LINC00976.2	-29065	MYC	TSS +1261Kb
268	Uveal	chr8	128993987	000125947	000125947_128993987	LINC00976	LINC00976.1	-28057	MYC	TSS +1259Kb
281	Uveal	chr8	129316452	000125980	000125980_129316452	LINC00977	LINC00977.2	-75517	MYC	TSS +1580Kb
308	Uveal	chr8	129685782	000126029	000126029_129685782	CCDC26	CCDC26.3	-5578	MYC	TSS +1948Kb
319	Uveal	chr8	129687031	000126029	000126029_129687031	CCDC26	CCDC26.4	-6787	MYC	TSS +1950Kb

FIGURE 3 for reviewers

The data in Figures 4-6 match up HiC data to CUT&TAG or ChIPseq data, and is a strength of the paper. It is not clear, however, how the peaks identified were matched to the HiC data. While the interpretation of these data by the investigators is logical and a likely possibility, it may be argued that these are distal enhancers for neighboring genes. There are some fitness scores in the figures, some clarification of how these were calculated and datasets compared would be beneficial.

We matched the Cut&Tag/ChIP-seq data with the Hi-C/Hi-ChIP data based on the genomic coordinates of peaks (Cut&Tag/ChIP) and tiles (Hi-C/Hi-ChIP) and aligning the coordinates of the selected loci for visualization purposes. Of note, we only report Hi-ChIP loops with significance FDR < 0.05 (now annotated in figure legends).

For the representation of the L2FC in figure 4A, 5A, 6A, S8A, S10C, we generate a bigwig track containing the Log2FC at day 22 for each sgRNA (compared to plasmid library representation) with a span of 250 bp just to enhance the visualization when representing large loci (a 20bp span would be too thin to be visualized). Also, this dataset was aligned with the genomic coordinates of the visualized loci. We updated figure legends for clarity.

The calculation of fitness scores in Figure 3A is described in the materials and methods section.

There are some different FDR cutoff ranges mentioned (ie, line 124, Figure S2, methods page 10, lines

367). It is not clear why these cutoff ranges were selected and why there is different FDR cutoffs between different cell lines. Some clarification in the methods is suggested.

We apologize for not having explained in detail our methodology for selection of significant YAP ChIP-seq peaks. Due to variability in ChIP target concentration in each cell line and ChIP efficiency, after standard peak calling with MACS2, by applying a standard FDR threshold to all 4 models, the contribution of peak representation from different models would be unbalanced. Thereby we decided to estimate a data-driven significance cutoff for each cell line model.

By plotting the distribution of FDR values of all peaks for each cell line, our goal was to retrieve the peaks that fell in the right tail of the FDR distribution and confirmed that the cutoff would be different between models.

To identify the precise cutoff, we plotted the histogram (as an empirical cumulative distribution function) and fitted a curve (function smoothed.spline()). The second derivative of the fitted curve is the inflection point and our desired threshold, obtained empirically from the data (see FIGURE 4 for reviewers below). These data can be added in a supplementary figure if deemed necessary by the reviewer.

While this was an interesting exercise to identify the most robust YAP peaks in each cell line, due to the limitation of the number of sgRNAs that can constitute a CRISPR library, we arbitrarily selected only sgRNAs targeting summits with an FDR cutoff $< 10^{-49}$. Thereby the initial filtering of YAP peaks would have minor impact on the library design. For completeness we update the methods to incorporate the description of YAP peak selection.

FIGURE 4 for reviewers

There is some concern for overgeneralization, such as in conclusions on lineage specificity based on analysis of only two, and often only one, cell type of each cancer group. Further, the schematic shown in Figure 6 involving lineage specific transcriptional groups is very interesting, but may be a stretch based on the data presented here. However, concerns here are reduced due to the overall comprehensive experimental approach employed throughout the manuscript.

We do acknowledge the limitations of using a small number of cell lines to draw conclusions on cancer lineages and we appreciate that the reviewer values our multi-omics approach to de-risk some of the conclusions. Nevertheless, we extended our validation strategy for lineage-specific YRE hits (see FIGURE 2 for reviewers above and FIGURE 10-14 for reviewers below) and for the data related to sensitivity to TEAD inhibitor, we validated our findings in additional cellular models (see FIGURE 5 for reviewers below and the same data have been added to the manuscript in Figure 1B) in line with the currently ongoing clinical trials for TEAD inhibitors which are limited to indications displaying canonical Hippo pathway aberrations such as Mesothelioma.

FIGURE 5 for reviewers

Minor points

There are some minor typographical errors in the document, including "T systematically interrogate (the) Yap cistrome" (line 36), "released into (the) nucleus" (line 54), and "derived network of melanocytic transcription factors" (line 82). Overall, the manuscript is well written and with enough scientific description in most of the document.

Thank you for pointing out these typos. We fixed in the revised version.

There are some formatting errors with references, such as Kaya-Okur et al 2019 (lines 443, 500).

We fixed these issues in the revised manuscript.

The link listed in line 536 is no longer an active website and leads to a 404 page.

We apologize for that. We replaced the old link with the new github link from Arima https://github.com/ArimaGenomics/mapping_pipeline

Comment

This work brings up several very interesting scientific questions. For example, are YAP/TAZ involved directly in lineage specificity, or are recruited by lineage specifying factors and promote or modulate the expression of these genes? This is an interesting potential following the schematic presented in Figure 6. What these factors are specifically doing in terms of gene expression is not understood, and this work provides some clues.

We thank the reviewer once again for appreciating our manuscript and we do agree that our manuscript raises several follow up questions. One of the key aspects that we are investigating is how YAP can be recruited to specific genomic sites according to cell states and the detailed molecular mechanisms triggered by the disruption of the interaction between YAP with TEADs or other TFs.

Reviewer #2 (Remarks to the Author): expert in YAP/Hippo signalling

Barbosa et al. present a detailed and functional analysis of YAP target genes in two cancer types that are driven by YAP activation, mesothelioma and uveal melanoma. They found that the target gene programs are different between the two cancer types. Surprisingly, they found that YAP-TEAD interaction is required only in mesothelioma but not uveal melanoma, indicating that YAP acts through different mechanisms in the two cancer types. The study is well executed and the conclusions largely follow from the data. The results are important as they inform therapeutic strategies and potential use of TEAD inhibitors to treat cancer. However, the authors should add analyses of YAP target gene engagement under TEAD inhibitor treatment and combined TEADi and YAP knockdown to fully support their conclusions.

We thank the reviewer for appreciating our study and particularly the impact of our findings. See below our efforts to strengthen the connection between the functional role of YAP and TEAD inhibition.

1. Because TEADs can act as suppressors in the absence of YAP, the authors need to perform a combination treatment of TEADi and YAP knockdown in order to show that the effects of YAP knockdown in uveal melanoma do not require TEAD. Without this experiment the results are not conclusive.

We thank the reviewer for proposing this intriguing theory. Accordingly, we performed dose response curve to evaluate the sensitivity of 92.1 uveal melanoma cells and NCI-H2052 mesothelioma cells to TEADi upon silencing of YAP or a pan-lethal gene such as PLK1. The results reported below demonstrate that, while the genetic knockdown leads to the expected sensitivities, the addition of TEAD inhibitor does not display an additive effect. See below FIGURE 6 for reviewers, these data can be added to the manuscript at discretion of the reviewer.

FIGURE 6 for reviewers

2. Do YAP target sites have TEAD motifs and are they bound by TEADs (cut&tag)? This is especially important to know for the lineage-specific sites.

We apologize for not having presented the TEAD occupancy and motif occurrence at lineage-specific sites. This is indeed an interesting point since both genetic and pharmacological tools demonstrate that TEAD is functionally dispensable in Uveal Melanoma cells, despite TEAD genomic occupancy at lineage-specific sites. To validate the latter, we computed coverage of ChIP-seq signal obtained with a TEAD4 or panTEAD antibody in the common and lineage-specific hits reported in figure 3C. We do observe enrichment (relative to input) for both panTEAD and TEAD4 at all scoring sites with a slight increase in signal in Common and lineage-specific sites according to the cell line tested (MSTO211H and NCI-H2052 for Mesothelioma vs. 92.1 and OMM1 for Uveal Melanoma) according to the metapeak plots on top of each heatmap in FIGURE 7 for reviewers below. These data could be added to a supplementary figure of the manuscript upon request of the reviewer.

FIGURE 7 for reviewers

We additionally computed local motif enrichment analysis in a window of +/- 250bp from YAP peak summit in lineage specific sites using CentriMO (from Meme suite, Baley et al, NAR, 2012, see FIGURE 8 for reviewers below). We observed enrichment for TEAD motif (in red) around YAP peaks summit in both Mesothelioma and Uveal melanoma specific sites. We additionally observe an enrichment for motifs recognized by MAPK TFs (green shades) in both peaksets, however such enrichment is sharper in Mesothelioma-specific YREs and shifted compared to TEAD motifs as previously reported (Zanconato et al., Nature Cell Bio, 2015), while in uveal melanoma the “MAPK motifs” are more broadly distributed in a window of ~240 bp around YAP peaks summit. In uveal melanoma specific sites we detected specific enrichment for the motifs of TFEB and MITF as reported in our analysis in current figure 2D.

Top Motif for TEADs or MAPK TFs is displayed for clarity of the image

3. What is the effect of TEAD_i on YAP occupancy at YREs? I am surprised that the authors did not include such analysis as it would answer one of the most straight forward questions resulting from their model that YAP can act independent of TEAD in uveal melanoma.

RESPONSE: REDACTED

Minor comments

1. line 133: the two library pools are not well described

We added information about the two library pools in the main text to describe the information displayed in Figure S2A.

2. How far on the DNA does KRABi exhibit inhibitory effects and how does this affect the conclusions?

As previously reported (Hsu, et al., *Nature Methods* 2018, PMID 30504875), the dCas9-KRAB (CRISPRi) system starts perturbing functional elements up to ~500bp distance. The rationale behind our choice for dCas9-KRAB lies on the ability to design multiple sgRNAs for each YRE rather than relying on regular DNA-cutting Cas9, for which sgRNAs would have to be designed precisely on the TF motifs contained in the YREs. We believe that this strategy adds statistical robustness to the results of our screens, while prompting us to add several layers of multiomic datasets to appropriately correlate sites of cell line-specific YAP binding and associated histone marks to sites reducing cellular fitness upon KRAB recruitment.

3. line 282: “silencing” is not the correct term here. Better “activity” or rephrase.

We corrected the term according to the suggestion

Reviewer #3 (Remarks to the Author): expertise in CRISPR screening methodology

In their manuscript entitled “Cancer lineage-specific regulation of YAP responsive elements revealed through large-scale functional epigenomic screens”, Barbosa et al. study functional differences downstream of YAP in two distinct cancer types. They choose malignant pleural mesothelioma (MPM) and uveal melanoma (UM) as prototypes for driver mutations in/outside the canonical Hippo pathway as well as TEAD dependency/independency respectively.

To identify relevant target genes downstream of YAP in both cancer types, the authors perform a CRISPRi screen recruiting a dCas9-KRAB fusion to YAP binding sites identified in a ChIP experiment. These screens and validation thereof is well-executed and delivers high confidence target loci with differential fitness effects upon KRAB recruitment.

The authors perform an impressive amount of epigenetic analyses to map out binding sites of YAP, TAZ, TEAD, as well as various histone marks and also analyze chromatin structure by HiC. The manuscript further contains e.g. various experiments using chemical inhibitors, overall a true tour de force. They can clearly correlate sites of cell line specific YAP binding and associated histone marks to sites reducing cellular fitness upon KRAB recruitment. Further, they convincingly demonstrate proximity of these regions to the transcriptional start sites of respective genes by HiC. As example, they identify cell type specific enhancer elements at the MYC locus for the studied cell lines. (Of note, in Figure S7B only one of the two +1800 sgRNA in the validation scores in 92.1 at a time scale that would be expected for the MYC locus and effects on MYC expression are overall small. Repeating this experiment with further sgRNAs scoring in the initial screen would thus increase trust in the result.)

We thank the reviewer for appreciating our study and our strategy to characterize functional YREs. Regarding the validation data on lineage-specific YREs in MYC locus, following the suggestion of the reviewers (see also comments from Reviewer 1), we extended the analysis to two cell lines per lineage using multiple sgRNAs, and measured both phenotypic effects and MYC expression levels. We believe that we could now further corroborate our findings that different YREs (+440 Kb in Mesothelioma cells and +1800 Kb in Uveal Melanoma cells) are able to regulate MYC expression and proliferation in specific lineages. These data in FIGURE 10 for reviewers below are also included in a revised figure S7 B-C.

FIGURE 10 for reviewers

The authors go on to demonstrate lineage specific cooperativity with other signaling pathways for MPM and UM respectively. While the MAPK responsive TFs JUN, FOSL1 and FOSB appear to be specifically required in MPM and silenced upon recruitment of KRAB, MITF, SOX10 and PAX3 are only essential in UM. As for differentially active enhancers in the MYC locus such result does not come at great surprise given known genetic dependencies and mechanisms of e.g. resistance to targeted therapies. In these sections about differential interactions with other TFs however data display becomes incomplete and inconsistent. The authors refer to Fig. 3C for lineage specificity, however the resolution here does not allow for full evaluation. Fig 5A, 6A, S8A, and S10C lacks tracks for the KD experiment of the other cell lines respectively. Moreover, validations by competition assay are done only selectively (e.g. for FOSL1 in S9A), and expression analysis is also not consistent with competition assays but rather only displayed in responder cell lines (S9B, S10B). Please provide a more coherent validation and support of your claims based on the methods well-established in your labs.

We thank the reviewer for this comment and apologize for not providing a comprehensive validation of our findings in the first version of the manuscript (indeed similar points were raised by reviewer 1). Regarding figure 3C, we provide a detailed Supplementary Table 8 containing extensive information about each region reported in the heatmap in Figure 3C. Regarding genome browser tracks, for the mesothelioma-specific sites, we observe that the same regions (green vertical stripes) are largely inactive in uveal melanoma cells (not scoring in the YMC screen, low/absent H3K27ac or low/absent YAP ChIP-seq signal) as depicted in FIGURE 11 for reviewers below.

FIGURE 11 for reviewers

The corresponding reciprocal observations were done for the uveal melanoma-specific sites in mesothelioma models, as shown in FIGURE 12 for reviewers below.

FIGURE 12 for reviewers

We additionally extended the validation package to include competition assay in all four cell lines employed in the study, additional sgRNAs for JUN, FOSB and FOSL1 YREs, and measured also corresponding target gene repression as captured in FIGURE 13 for reviewers and reported now in Figure S9 A-E. Of note, FOSL1 expression was not detected in uveal melanoma cells and therefore those data are not plotted.

FIGURE 13 for reviewers

We performed similar analyses for uveal melanoma-specific hits (see FIGURE 14 for reviewers) and again validated the selectivity of lineage-specific hits. Note that PAX3 and MITF are not expressed in mesothelioma cells, hence those data are omitted from the plots (we show the data for SOX10 since we also demonstrate the lack of modulation of the POLR2F gene which hosts the SOX10 gene in an intron). All these data have been added in the manuscript as supplementary S10 A-F.

FIGURE 14 for reviewers

The authors start the analysis based on their mapping of YAP binding sites to then correlate epigenetic features to the ability to silence with CRISPRi. Their studies culminate in a model (Fig. 6C) whereby YAP together with lineage specific transcription factors induces target genes and thus recruitment of KRAB to that locus results in reduced cellular fitness. However, the authors only show that KRAB reduces fitness if bound to these sites, so in other words they show these sites to be accessible and likely functional enhancers. They provide no functional evidence that YAP indeed induces expression in said manner and at that locus, it may just bind the open enhancer in a cell type specific manner. While I agree that the correlative evidence is strong, functional claims are not really supported by data. An alternative and plausible interpretation of the data could e.g. be that Cas9 binding/KRAB is particularly effective if recruited to accessible loci (ChIP) in proximity (HiC data) to the TSS.

RESPONSE: REDACTED

In this context I may mention that I do not find figures 5B and 6B particularly convincing, as the data are strongly overlapping between the three groups and minimal differences could be explained by enhancer accessibility.

We apologize for the lack of clarity on those two figures. We realize that those panels were crowded potentially impacting the visual appreciation of the differences between groups. Thereby we re-made the violin plots by keeping the default trimming parameter of `geom_violin` in `ggplot` and plotting only the median value rather than the 0.25/0.5/0.75 quartiles lines. Also, we adopted a star-code for p-values (* $P < 0.05$; ** $P < 0.01$; *** $P < 0.001$, and not significant (n.s) $P > 0.05$), hopefully allowing a better appreciation of the data plotted.

If the authors wish to maintain the claim visualized in the model, functional data are required at least for some examples. Various experiments could support the claim functionally such as ChIP or HiC upon acute loss of YAP, acute overexpression, or recruitment of YAP to lineage specific enhancers e.g. via dCas9. Locus specific degradation of YAP could also be envisioned. Alternatively, YAP binding motifs within identified regions could be targeted with WT Cas9 or better by induction of point mutations or HR to reach a higher resolution in the genetic perturbation and confirm results independent of KRAB. (In this case care must be taken that a transient DNA damage response does not result in false positive results.) I would consider such functional experiments critical in light of the claims of the manuscript, but would leave the decision of how to probe functionality to the authors.

RESPONSE: REDACTED

The figures are of remarkable good quality and offer a very consistent layout, which I would like to point out as an outstanding feature of this manuscript, congratulations! Similarly, the manuscript is well structured, very well written, and overall very mature.

We thank once again the reviewer for appreciating our manuscript and we're glad that our visualizations and overall delivery was clear and well received.

Reviewer #4 (Remarks to the Author): expertise in melanoma epigenetics

In this study, the authors employed large-scale functional epigenomic screens of YAP regulatory elements (YRE) in both MPM and UM, and revealed unanticipated lineage-specific features of the YAP regulatory network that provide important insights to guide the design of tailored therapeutic strategies to inhibit YAP signaling across different cancer types. The manuscript is well-organized, well-written, and more valuable, generating a large amount of functional screening data. The reviewer has several suggestions:

1. The authors focused on cancer-specific patterns, especially in MPM and UM, for example, figure 1B – is it possible to examine the pattern in other cell lines of different cancer types? This may be completed through public datasets, such as GDSC and CTRP, which cover thousands of cell lines with hundreds of drugs – if the TEADi is included in the drug screening.

We do appreciate that large scale datasets could well complement our data. Unfortunately, TEAD inhibitors are not part of large-scale drug screening datasets (GDSC or CTRP), thereby we decided to expand our original panel of TEADi treatments to other cellular models:

- Four mesothelioma cell lines (MSTO-211H, NCI-H2052, H226 and Acc-Meso1)
- Four uveal melanoma cell lines (92.1, MEL202, OMM1 and MP41)
- Other cell lines – cutaneous melanoma (MeWO), lung squamous cell carcinoma (LK2), hepatocellular carcinoma (SNU499) and pancreatic ductal adenocarcinoma (AsPc1).

The results (see FIGURE 4 for reviewers above as commented for Reviewer 1 and updated Figure 1B) support the current rationale behind ongoing clinical trials for TEAD inhibitors, which are restricted to indications displaying canonical Hippo pathway aberrations, such as Mesothelioma.

2. Similar to the above point, the authors may also consider expanding the scope by utilizing public data, such as DepMap for CRISPR screening result in different cancer cell lines, if applicable.

We thank the reviewer for the suggestion. While our paper highlights phenotypic differences between genetic silencing of YAP and TEADi treatments, large scale datasets of TEADi are lacking (see above) and additionally DEPMAP data suffers from low representation of cell lines from melanoma within the eye lineage. Indeed out of the 15 Uveal Melanoma cell lines screened in the latest version of DEPMAP (22Q4), 2 are retinoblastoma cell lines, 5 are non cancerous models and only 3 bear high confidence mutations in GNAQ/GNA11 and are sensitive to these oncogenic drivers.

3. The authors generate a large amount of data, which will be valuable resources for the research community. The authors should release the related data, not only the HiC and Hi-ChIP (deposited in GEO but without the GSE number) but other data, including pooled CRISPR screening data, CHIP-seq, and Cut& Tag data.

We apologize for not having deposited earlier our NGS data to a public depository. Currently all our data (n= 99 samples) have been deposited to SRA with BioProject ID: PRJNA949402.

Data can be accessed by reviewers following this link:

<https://dataview.ncbi.nlm.nih.gov/object/PRJNA949402?reviewer=2t64btget8d4ddq58a6v5u5f6p>

Data will be publicly released upon manuscript acceptance.

Reviewer #5 (Remarks to the Author): expertise in HiC and Cut&Tag analysis

In this manuscript, Barbosa et al. present a study of the YAP transcriptional co-activator in the context of two cancers: malignant pleural mesothelioma (MPM) is driven by mutations in components of the Hippo signaling pathway, which is responsible for suppressing the nuclear activation of YAP, while in uveal melanoma (UM), YAP is activated by oncogenic mutations in GNAQ/GNA11 GTPases, independent of the Hippo pathway. The authors utilize genome-wide functional epigenomic screens to reveal YAP-responsive elements (YREs) shared by and specific to each cancer lineage.

A comprehensive analysis is performed here utilizing a wide-range of assays including CRISPR-based YMCi libraries, ChIP-Seq of an array of epigenetic marks and HiC/HiChIP to look at chromosome conformation. Using this platform, the authors systematically interrogate YAP-responsive cis-regulatory elements in a genome-wide fashion and present novel insights about lineage-specific mechanisms involved in YAP-dependent cancers. Their findings suggest shared and lineage-specific programs operating in the two cancers, which provide guidance for development of selective therapeutic strategies in YAP-related cancers.

I believe this is an excellent study and the authors have done a great job summarizing their findings succinctly and providing suitable context for the reader. I have some minor comments predominantly regarding data visualization.

Specific comments:

1. Page 4, line 124: It is not clear from the main text, how the top 18063 selected YAP-peaks are distributed in terms of lineage-specificity. This information should be highlighted here instead of just in supplementary figures.

We updated the text from the referred passage with detailed description of peak distribution across lineages and cell lines:

“Next, we designed a comprehensive CRISPR library containing approximately 160'000 sgRNAs targeting the most significant YAP-peak summits among the four cell lines ($n_{\text{total peaks}}=18039$, $-\text{Log}_{10}\text{FDR}>49$; 2941 peaks common to both lineages, 6389 mesothelioma-specific peaks, 2119 uveal melanoma-specific peaks, 2938 cell line-specific peaks and 3652 peaks in other distributions) (Figure S2A-S2C and Supplementary Table 1).”

Of note, in the first version of the manuscript the number of YAP-peaks targeted was mistakenly reported as 18063, however the correct number is 18039. We apologize for this mistake.

2. Figure 2B: This figure is hard to understand. The spread of points in pan-lethal and experimental groups look very similar, but the violin plots are very different. Why is this?

We thank the reviewer for offering the opportunity to clarify this point. The “pan-lethal” (PL) sgRNA control group contains 1030 sgRNAs (see figure S2A for details), targeting essential genes for cell survival (defined in Horlbeck et al. eLife, 2016). Therefore, we expect the representation of the majority of sgRNAs within this group to decrease overtime, translating into a median L2FC lower than 0, and creating an elongated density plot, where very few sgRNAs remain around L2FC 0.

On the other hand, the “experimental” sgRNA test group contains approximately 160000 sgRNAs, of which a large fraction targets enhancer regions, and only a portion is expected to have an impact on survival and proliferation of the probed cell lines. For this reason, the L2FC of most sgRNAs will be around 0, creating a high density of dots (and overplotting using the jitter plot) around this value, hence the wider violin plot can help appreciating the data distribution. We therefore used a grey colored scale to highlight the relatively small fraction of sgRNAs that significantly depletes overtime in the “experimental” sgRNA group.

Also, how are ‘experimental’ groups defined? Are these all 18000 selected regions? I could not find this information in figure legends or in methods.

We apologize for having omitted this information earlier. Figure 2B was built using single sgRNA data values and, in the particular case of the “experimental” group, data from all sgRNAs (approximately 160000) targeting the ≈18000 YAP peaks were used. Figures S2A-C contain details about the design of the “control” and “experimental” sgRNA groups. Minor changes were made to figures S2A and S2B to use a consistent nomenclature and highlight total number of “experimental” sgRNAs and number of YAP peaks targeted (see below red arrows in FIGURE 17 for reviewers).

FIGURE 17 for reviewers

We also added the following description to the material and methods’ section **Design of YMCi-160K and YMCi-13K libraries**:

“Overall, the design of the sgRNA library targeting YAP sites followed multiple steps. First, we selected the most robust YAP ChIP-seq peaks for each cell line and created a consensus peak-set. Next, we identified sgRNAs targeting such peaks with defined criteria. Finally, due to constrains in the number of sgRNAs that could be incorporated in libraries, we further restricted to guide RNAs which targeted the peak summits with the highest significance ($n_{sgRNAs} = 159612$; $n_{total\ peaks} = 18039$).”

3. Page 4, line 143-146: “Within experimental sgRNAs, as previously reported for Estrogen Receptor, only a small fraction of perturbations displayed significant scoring both in UM and MPM, suggesting that only a subset of YREs is necessary for cell proliferation (Figure 2B).”

It is very difficult for the reader to make this conclusion from Figure 2B. Either the figure legend needs to be more descriptive, or this sentence needs to include more information.

We apologize for not making this information visually accessible to viewers. Figure 2B is indeed a composite of multiple plots:

- an inner box plot representing the median L2FC, which is around 0 for experimental guides;
- a side kernel density plot (aka violin plot) to appreciate the full distribution of the data around that value;
- a jitter plot depicting the outliers, shown with their significance color-coded in shades of grey.

Given the high number of sgRNAs belonging to the experimental group, we highlighted in darker grey the small number of sgRNAs significantly depleting ($L2FC < -1$, $FDR < 0.05$) at day 22 in H2052 ($n=896$) or 92.1 ($n=2235$). In legend of Figure 2B, we added the information about group size (nr of sgRNAs) for each category. We can also provide details about number of sgRNAs significantly depleted in each group at different time points, if the reviewer finds that advisable (see in FIGURE 18 for reviewers below an exemplificative graph for the “experimental” sgRNA group at day 22).

FIGURE 18 for reviewers

4. Page 4, line 149: Even though the authors analyzed three different time-points: 8, 15 & 22 days post-infection, only day 22 is analyzed here. The authors should explicitly note the reason behind this choice here.

We apologize for not explicitly explaining why we plotted only day 22 for the comparison of NCI-H2052 vs. 92.1 primary screen. Fundamentally there was no specific reason besides the fact that at day 22 we see the maximal effect/differences between the two cell lines, similar to what has been observed for the individual analyses for each model shown in figure 2B. See in FIGURE 19 for reviewers below the analyses at the three timepoints. We are happy to add the plots for additional timepoints to supplementary S4D for completeness.

FIGURE 19 for reviewers

5. Figure 4B: The ‘UM-specific’ +1.8Mb region appears to have strong HiC signal in the MPM cell line, suggesting physical interactions with the MYC locus. There also appears to be some H3K4me1 CTCF signal at this locus in this cell line. Even though there is no H3K27Ac signal, the above appear to indicate

that the structure of this particular region is not lineage-specific. The reverse is not true of the UM-cell line. How do the authors explain this?

We thank the reviewer for this interesting observation. Indeed, as evident by H3K27ac Hi-ChIP it seems that the distal region (around +1.8-2 MB) of the MYC locus is involved in contacts with the MYC TSS in both cell lines, while the proximal region of the locus is somehow “isolated” in a smaller mesothelioma-specific loop (around +440 Kb) in mesothelioma cells. To better appreciate quantitative differences in interaction frequency between cell types, we report in FIGURE 20 for reviewers below the delta Hi-C signal (20Kb resolution Mesothelioma / Uveal melanoma signal) which confirms the specificity of the mesothelioma signal at +440 Kb and a “spread” enriched signal in uveal melanoma cells for the second part of the MYC locus presented in the maps.

FIGURE 20 for reviewers

We also made minor changes to Figure 4 of the manuscript, in order to more accurately depict the UM-specific region – we changed the width of the purple-shaded area to encompass the second YAP peak only (and the one which in fact scores in our uveal melanoma YMCi screens), excluding the one immediately before which is also present in the NCI-H2052 mesothelioma cell line and has the referred H3K4me1 signal.

6. Page 6, line 209-211: This claim is not very well supported by the figure. The differences appear to be varied and it's hard to collate the p-values from all the different groups shown in the figure. The authors should consider depicting this information using a table.

We thank the reviewer for the suggestion to improve the visualization and interpretation of these data. We replaced the p-value numbers for asterisk-coded significance (*P < 0.05; **P < 0.01; ***P < 0.001, and not significant (n.s) P > 0.05). Figure legends were updated with this information.

The source data of the violin plots shown in Figure 5B and 6B can be found in the tables below (FIGURE 21 for reviewers), containing the following information:

- Median coverage of each factor profiled by Cut&Tag;
- Significance values (p-value) for each group comparison;
- Asterisk-code adopted for each p-value;

The information contained in these tables can also be provided as supplementary data, if deemed necessary by the reviewer.

Figure 5B:

Figure 6B:

FIGURE 21 for reviewers

7. Figure 5C: What is the meaning of the 'score: ...' numbers at the top right of the figure. This is not described in the figure legend.

We thank the reviewer for pointing out the missing information. The numbers on top right of the heatmaps correspond to the “Loewe Synergy Score” calculated for drug combinations. These scores can be interpreted as the average excess response due to drug interactions (i.e synergy score of 20 corresponds to 20% of response beyond expectation). Since combination synergy can be highly context-specific, there is no particular threshold to define a good synergy score, but scores higher than 10 signal that the interaction between two drugs is likely “synergistic”, while likely “additive” if lower than 10.

8. Correlation heatmaps depicting positive values ranging between 0 and 1 are shown using divergent colormaps, e.g. Fig 1C, S1B, S5C, S9C. This type of colormap is visually misleading as values in the middle of the scale are shown in white and effectively highlighted, while values at the bottom of the scale (with 0 correlation) also appear unduly strong. These figures should really use a sequential colormap, e.g. viridis.

We updated all heatmaps with viridis coloring, according to the reviewer’s suggestion. The new heatmaps can be found in Figure 1C, Figure 1E, Figure S1B, Figure S5C and Figure S9F.

Minor comments:

1. Page 8, Line 274: AP-1 motifs “flanking” TEAD motifs

The typo has been corrected.

REVIEWERS' COMMENTS

Reviewer #1 Remarks to the Author

This is a review of revised manuscript NCOMMS-22-33613A "Cancer lineage-specific regulation of YAP responsive elements revealed through large-scale functional epigenomic screens" by Barbosa, Galli, and colleagues.

This is a impactful study examining cancer lineage specific regulation of YAP on genomic enhancers, using a comprehensive epigenomic analyses. This work is strong both conceptually and technically. Conceptually, there is a broad impact, both on how different tumor types may respond to YAP/TAZ based therapeutics and on how the canonical HIPPO pathway functions in normal cell lineages. While other studies have examined YAP cistromics, here there are distinct patterns between two representative cancer types of different lineages. Technically, the research strategy is highly comprehensive and elegantly designed, with well-controlled experimental groups, bioinformatics, statistics, and multiple supportive methodologies. In several places in the manuscript, findings are validated through other approaches.

In this revised version of the manuscript, all previous points have been well addressed. This work is highly impactful and an excellent candidate for publication in Nature Communications.

Reviewer #2 Remarks to the Author

The authors have answered my questions.

Reviewer #3 Remarks to the Author

The authors addressed the requested changes to the manuscript in a comprehensive and convincing way and thus further improved upon an already good manuscript. They provide extra data corroborating the presented results.

The only issue that did not get fully addressed is the one regarding the functional role of KRAB at the enhancers as also brought up by me as well as reviewer 1 in point 3. It is well addressed with the confidential dataset on NVS TEAD_i presented to the reviewers, yet these data will not be part of the manuscript.

I specifically concur that inclusion of these data would be beyond the scope of this study and should not be insisted on, nevertheless in absence thereof the functional claims as well as the point raised by reviewer 1 remain poorly supported. A sentence on the limitations of this study might therefore be warranted in the discussion, but I leave this decision to the editor.

Congratulations to this insightful and important study!

Reviewer #5 Remarks to the Author

I would like to thank the authors for addressing all my concerns in their revision. I have no more comments.

RESPONSE TO REVIEWERS' COMMENTS

Reviewer #1 (Remarks to the Author):

This is a review of revised manuscript NCOMMS-22-33613A "Cancer lineage-specific regulation of YAP responsive elements revealed through large-scale functional epigenomic screens" by Barbosa, Galli, and colleagues.

This is a impactful study examining cancer lineage specific regulation of YAP on genomic enhancers, using a comprehensive epigenomic analyses. This work is strong both conceptually and technically. Conceptually, there is a broad impact, both on how different tumor types may respond to YAP/TAZ based therapeutics and on how the canonical HIPPO pathway functions in normal cell lineages. While other studies have examined YAP cistromics, here there are distinct patterns between two representative cancer types of different lineages. Technically, the research strategy is highly comprehensive and elegantly designed, with well-controlled experimental groups, bioinformatics, statistics, and multiple supportive methodologies. In several places in the manuscript, findings are validated through other approaches.

In this revised version of the manuscript, all previous points have been well addressed. This work is highly impactful and an excellent candidate for publication in Nature Communications.

We thank the reviewer for the insightful comments that strengthened the findings of our manuscript

Reviewer #2 (Remarks to the Author):

The authors have answered my questions.

We thank the reviewer for prompting us to test/validate interesting hypothesis and appreciating our efforts during the revision process.

Reviewer #3 (Remarks to the Author):

The authors addressed the requested changes to the manuscript in a comprehensive and convincing way and thus further improved upon an already good manuscript. They provide extra data corroborating the presented results.

The only issue that did not get fully addressed is the one regarding the functional role of KRAB at the enhancers as also brought up by me as well as reviewer 1 in point 3. It is well addressed with the confidential dataset on NVS TEAD_i presented to the reviewers, yet these data will not

be part of the manuscript.

I specifically concur that inclusion of these data would be beyond the scope of this study and should not be insisted on, nevertheless in absence thereof the functional claims as well as the point raised by reviewer 1 remain poorly supported. A sentence on the limitations of this study might therefore be warranted in the discussion, but I leave this decision to the editor.

Congratulations to this insightful and important study!

We thank the reviewer for the insightful comments and appreciating our efforts in addressing the points raised.

We are glad that the reviewer agrees with us that the use of a potent and selective acknowledge YAP/TAZ-TEAD PPI inhibitor could address the concern related to the use of the dCas9-KRAB system. We do concur that it is quite unfortunate that we cannot disclose in the current manuscript the data on our inhibitor, and we do agree to highlight the limitation of our study in a sentence in the discussion.

Reviewer #5 (Remarks to the Author):

I would like to thank the authors for addressing all my concerns in their revision. I have no more comments.

We thank the reviewer for the good suggestion to improve the quality of our manuscript.